

# Description of the resolution hierarchy of the global coupled HadGEM3-GC3.1 model as used in CMIP6 HighResMIP experiments

Malcolm J. Roberts[1], Alex Baker[2], Ed W. Blockley[1], Daley Calvert[1], Andrew Coward[3], Helene T. Hewitt[1], Laura C. Jackson[1], Till Kuhlbrodt[2], Pierre Mathiot[1], Christopher D. Roberts[4], Reinhard
Schiemann[2], Jon Seddon[1], Benoît Vannière[2], Pier Luigi Vidale[2]

[1]Met Office Hadley Centre, Exeter, U.K.
[2]National Centre for Atmospheric Science (NCAS), University of Reading, Reading, U.K.
[3]National Oceanography Centre, Southampton, U.K.
[4]European Centre for Medium Range Weather Forecasting (ECMWF), Reading, U.K.

*Correspondence to*: Malcolm J. Roberts (malcolm.roberts@metoffice.gov.uk)

**Abstract**. CMIP6 HighResMIP is a new experimental design for global climate model simulations that aims to assess the impact of model horizontal resolution on climate simulation fidelity. We describe a hierarchy of global coupled model resolutions based on the HadGEM3-GC3.1 model that range from an atmosphere-ocean resolution of 130 km-1° to 25 km-1/12°, all using the same forcings and initial conditions. In order to make such high resolution simulations possible, the

experiments have a short 30 year spinup, followed by at least century-long simulations with both constant forcing (to assess drift and the focus of this work), and historic forcing.

We assess the change in model biases as a function of both atmosphere and ocean resolution, together with the effectiveness and robustness of this new experimental design. We find reductions in the biases in top of atmosphere radiation components and cloud forcing. There are significant reductions in some common surface climate model biases as resolution is increased,

particularly in the Atlantic for sea surface temperature and precipitation, primarily driven by increased ocean resolution. There is also a reduction in drift from the initial conditions both at the surface and in the deeper ocean at higher resolution. Using an eddy-present and eddy-rich ocean resolution enhances the strength of the North Atlantic ocean circulation (boundary currents, overturning circulation and heat transports), while an eddy-present ocean resolution has a considerably reduced Antarctic Circumpolar Current strength. All models have a reasonable representation of El Nino - Southern Oscillation. In general the

biases present after 30 years of simulations do not change character markedly over longer timescales, justifying the experimental design.

## 1 Introduction

There is now considerable evidence that enhancing model horizontal resolution can help to reduce systematic and long-standing climate model biases (Kinter et al. 2013; Small et al. 2014; Griffies et al. 2015; Roberts et al. 2016; Hewitt et al.

2016; Roberts CD et al, 2018; Roberts M et al., 2018), and hence potentially improving the robustness and trust in future





projections. Some of the evidence for this comes from previous Coupled Model Intercomparison Project (CMIP) exercises (Meehl et al. 2000, 2007; Taylor et al. 2012). However it can be difficult to assess the impact of horizontal resolution changes alone, as even when the same model is submitted to CMIP with multiple resolutions (relatively rare for coupled models), there are generally additional model differences. These may include retuning via parameter changes, and difficulty in assessing

model evolution due to extra complexity (e.g. interactive aerosol schemes, Earth System components).

CMIP6 HighResMIP (Haarsma et al. 2016) is a new experimental design that specifically focuses on assessing the impact of increased horizontal resolution on mean state biases, using model configurations designed for this purpose. The protocol encourages minimal model changes as resolution is increased, the use of a common, simplified aerosol optical properties scheme (MACv2-SP; Stevens et al. 2017), as well as common initial conditions and other standard CMIP6 forcings (Eyring et

al. 2016). However due to the increased costs of such enhanced resolution models, some compromises need to be made. One such is the length of coupled model simulation - it is not affordable to execute long pre-industrial (PI) spin-up and PI control simulations (typically many 100's years) as used in the CMIP6 DECK simulations (Eyring et al. 2016).

Hence given a new protocol, together with model resolutions which range wider than those used in previous CMIP exercises, we need to assess the efficacy of this experimental design, finding its strengths and weaknesses, in addition to using it to assess

the impact of model horizontal resolution. Similar assessments with other global climate models are ongoing and can be found in Roberts CD et al. (2018), Cherchi et al. (2019), Voldoire et al. (2019), Gutjahr et al. (2019), Sidorenko et al. (2019), Haarsma et al. (in prep).

Here we describe the HadGEM3-GC3.1 model (Williams et al. 2017) as configured for HighResMIP, with a resolution hierarchy spanning from a standard CMIP-type resolution (130 km-1°, atmosphere-ocean, non-eddying ocean) via a 25 km

eddy-present ocean resolution, through to 25 km-1/12° and hence including an eddy-rich ocean. Our goals in this work are as follows:

1.    How does horizontal resolution impact on the simulated coupled climate, in particular on model biases, mean state and variability?

2.    How well does the CMIP6 HighResMIP protocol work in isolating these impacts?

3.    Are there areas in which it will be difficult to assess the models due to the protocol (e.g. drift bigger than signal, 100 years not long enough, not enough ensemble members for robust differences)?

4.    Do the longer control-1950 simulations shown here differ significantly from the initial 100 years, and do they reveal any further insights?

The focus of this work is on the spinup period and the control simulations (constant forcing), together with the longer-term

behaviour in several of the models. In section 2 we describe the model configuration at different resolutions, together with aspects of implementation of the HighResMIP experimental protocol. Results are shown in section 3 on the impact of resolution on aspects of the individual model components as well as the coupled evolution mean state and variability, and in section 4 we will summarise our experiences and discuss future work.





## 2 Model description

### 2.1 HadGEM3-GC3.1 used for CMIP6 and differences for use in HighResMIP

The configuration of the global coupled model HadGEM3-GC3.1 for submission to the CMIP6 DECK (Eyring et al. 2016) is described in Williams et al. (2017), Menary et al. (2018) and Kuhlbrodt et al. (2018). It incorporates a global atmosphere-land

configuration called GA/GL7.1 (Walters et al. 2019), with a new modal aerosol scheme (GLOMAP-mode; Mulcahy et al., 2018). The atmospheric model uses a regular latitude-longitude grid, and has 85 levels extending to 85 km. The global ocean component is called GO6 (Storkey et al. 2018), which uses the NEMO ocean model (Madec et al. 2016) at vn3.6, having a tripolar grid, with 75 ocean levels (and top level thickness of 1m). The sea ice model configuration is GSI8.1 (Ridley et al. 2018), which uses the CICE5.1 model (Hunke et al. 2015). Coupling between atmosphere and ocean models is performed by

the OASIS-MCT coupler (Valcke et al. 2015) with conservation for the heat and freshwater terms and with surface fluxes calculated on the atmosphere grid. The coupling period is set to one hour for all models.

The HighResMIP protocol recommends the use of the MACv2-SP scheme (Stevens et al. 2017) for simplified and standardised aerosol forcing. This specifies the change of anthropogenic aerosol optical properties over time, and hence enables easier comparison between different models, while retaining the model's own aerosol mean background climatology and hence

requiring little or no additional tuning. It is used here in place of the prognostic GLOMAP-mode scheme.

### 2.2 Model resolution differences

The different model resolutions used in this work, together with parameterisation and parameter choices, are summarised in Table 1. We will henceforth use the CMIP6 naming conventions and CMIP6 nominal resolutions when describing the models. The HadGEM3-GC3.1 model has very few parameter values explicitly changed as model resolution is varied (Table 1). In the

atmosphere, the only explicit parameter change (USSP launch factor) is used to produce a reasonable period for the Quasi-Biennial Oscillation (QBO), as described in Walters et al. (2019). For the ocean model, there are more changes, principally because we move from a regime at 100 km resolution (LL model) where the ocean mesoscale (ocean eddies, boundary currents) are strongly tied to parameterisations and the requirements of numerical stability, to eddying regimes at 25 km and 8 km where these properties are increasingly explicitly resolved. The parameter choices for the LL model are described in Kuhlbrodt et al.

(2018), with key differences from 100 km to 25 km resolution being the deactivation of the ocean eddy fluxes parameterization (Gent and McWilliams 1990), and the reduction of explicit dissipation parameters. The snow on sea ice albedo was adjusted to be lower in the LL model (by 2-3%, see Table 1) due to excessive sea ice thickness particularly in the Arctic (Kuhlbrodt et al. 2018).

In addition to explicit parameter differences, some model parameters and schemes are self-tuning, that is their controlling

parameters vary automatically based on model resolution. These include the stochastic physics schemes Stochastic Perturbation of Tendencies (SPT) and Stochastic Kinetic Energy Backscatter scheme (SKEB2), as described in Walters et al. (2019) and Sanchez et al. (2016).





Detailed descriptions of the atmospheric model differences, such as the impact of using MACv2-SP scheme, can be found in Vidale et al. (in prep), and further assessment of ocean model differences up to 8 km resolution are described in Storkey et al. (2018) and Mathiot et al. (in prep).

### 2.3 CMIP6 HighResMIP forcing

The simulations described here follow the CMIP6 HighResMIP protocol (Haarsma et al. 2016) in terms of the forcing datasets used. Aerosol forcing uses the MACv2-SP anthropogenic aerosol scheme (Stevens et al. 2017) combined with the model background mean (non-varying) natural aerosol, with model implementation described in Vidale et al. (in prep). The spinup-1950 and control-1950 experiments (Fig. 1) use mean values of the CMIP6 transient forcing datasets for aerosol (as above), solar (Mathes et al. 2017), ozone concentration (Hegglin et al. 2016), greenhouse gas (GHG) forcings (Meinshausen et al.

2016). Aerosol uses a monthly mean from the 1950-1959 period; ozone and solar use a monthly mean over 22 years centred about 1950 (in order to mean over the 11-year solar cycle); GHG uses the 1950 value of GHG global concentrations. The hist-1950 simulations (not analysed here) use the time-varying versions of these forcings.

2.4 Computational characteristics

Details of the computational performance of the models at different resolutions can be found in Vidale et al. (in prep). Table

2 shows an overview of the model costs and the volume of data output, primarily of the raw priority 1 diagnostics for the HighResMIP data request (Juckes et al. in prep). Since we expect the higher resolution models to represent weather processes and events in an improved way, there is a requirement for more high frequency output (1, 3, 6 hourly on multiple atmospheric levels, as well as daily ocean surface output) in order that we can assess these processes.

### 2.5 Model simulations

Most of the model data used in the following analysis is available from the CMIP6 Earth System Grid Federation, and can be located using the information in Roberts M (2018b, 2017a, 2017b, 2017c, 2017d) for resolutions LL, MM, HM, MH, HH respectively. Other model resolutions are not part of the official HadGEM3-GC3.1 CMIP6 HighResMIP submission, with the data available on request.

The analysis in this work uses data from the spinup-1950 and control-1950 HighResMIP experiments and hence is

representation of 1950 climatological conditions. It should be noted, when comparing these data to observations, that globally complete observational data are typically only available post-1970's, and hence one expects that some component of any "bias" will be attributable to this difference in the simulation and observational periods.

### 2.5.1 Spinup-1950 protocol

Given the expense of the higher resolution models, the typical spinup procedure as used in CMIP6 DECK simulations (many

100's years of pre-industrial (PI) spinup before piControl and historic simulations are initialised (Eyring et al. 2016) is simply not feasible here. Hence the HighResMIP protocol recommends 30-50 years using the spinup-1950 protocol from specified





ocean and atmosphere initial conditions - here we use 30 years. This "spun-up" state is then used to initialise the control-1950 and hist-1950 simulations, as illustrated in Fig. 1.

The common initial conditions for the ocean temperature and salinity are derived from the January 1950-54 mean of the EN4 ocean analysis (Good et al. 2013). This is bilinearly interpolated to the model ocean grid. For the higher ocean resolutions (25 km and 8 km), it was found that several days of simulation with a very short ocean timestep (typically one quarter of the standard value) was needed in order to remove small-scale instabilities introduced by the interpolation, particularly in the high Arctic and where the Mediterranean and Black Seas meet. The model was then restarted at 1950-01-01 with these derived ocean and sea ice initial conditions.

The atmosphere initial condition is derived from ERA-20C (Poli et al. 2016) in January 1950. Initial conditions for sea ice are taken from previous ocean-sea-ice simulations (at the same resolution) valid around 1979, since methods to initialise different sea ice models with common variables are less well-developed, and with the assumption that sea ice has a timescale of only several decades to quasi-equilibrate. The soil moisture has a relatively long memory, and its initial condition was taken from a previous HighResMIP atmosphere-only simulation using the same atmosphere resolution.

The spinup-1950 experiment with models using 25 km and 8 km ocean resolutions were only performed with one atmosphere resolution - MM and MH respectively - both using the 100 km atmosphere.

### 2.5.2 control-1950 protocol

The final state of all the model components at the end of spinup-1950 simulation are used to initialise both the control-1950 and hist-1950 experiments. As noted above, the MM spinup-1950 is used as initial condition of both MM and HM, and the MH spinup-1950 used for both MH and HH. This method seemed to work well for the ocean component, but for the soil moisture in the land component it was found that the more inhomogeneous soil properties at 50 km (HM, HH) resolution tended to retain less water. This led to a pulse of freshwater from land to ocean at the start of the control-1950 and hist-1950 simulations. To prevent this, the soil moisture from a previous 50 km simulation was inserted into initial conditions for the HM and HH simulations.

The control-1950 experiment is required to be at least 100 years in length, but can be much longer - it is the HighResMIP equivalent to the CMIP6 piControl. It uses the same forcings as spinup-1950. In this work, there are 1000 years of LL, 600 years of MM and 150 years of MH available for analysis.

### 2.5.3 hist-1950 protocol

These simulations have the same initial conditions as their respective control-1950 models. The forcings are now the time-varying CMIP6 forcings. Since we recognise that the models will continue to drift over time, due to the short spinup, we can use the difference between the hist-1950 and control-1950 simulations as an estimate of the impact of changing forcings on the climate state, assuming a common drift between the two simulation types. Analysis of these simulations is outside the scope of this work.





# 3 Results

## 3.1 Initial spin-up and radiative balance

The radiative balance of the different resolution models is shown in Fig. 2, in terms of the Top of Atmosphere (TOA) radiation, its outgoing shortwave and longwave (OSR, OLR) components, and the global mean surface temperature (ST), together with

observational estimates where possible. The TOA starts at between -1.5 to -1 W/m2 in the initial state of the models, and by the end of 30 years they have all adjusted to within ±0.3 W/m2, compared to an estimate of observed TOA in the range 0.23-0.6 W/m2 during 1985-2010 (Stephens et al. 2012; Wild et al. 2013; Allen et al. 2014). The mean TOA over the control-1950 period for each model is indicated by the box and whiskers plot to the right of Fig. 2. This radiative adjustment is accomplished in different ways by the different resolution models. The LL model has the largest changes in radiative terms with a sharply

increasing OSR by nearly 2 W/m2 over the initial 30 years (Fig. 2b), together with a reduction of 3.5 W/m2 in the OLR (Fig. 2c), both of which are deviations from the observed values. The latter is consistent with a 2 K decrease in the ST (Fig. 2d), while the former is consistent with an increase in Arctic sea-ice area (Fig. 5). During the control-1950 simulation, the surface temperature warms (Fig. 2d) and the OSR and OLR continue to adjust gradually over several hundred years. For comparison, as shown in Vidale et al. (in prep), the atmosphere-only simulations have a TOA which starts at around -1.5 in 1950, with

surface temperature of 288 K, OSR around 100-101, and OLR around 240.5-241.5 (which adjusts to around 239.5-240.5 at 1980 when the TOA is closer to zero).

The adjustment of the MM and MH models are reasonably similar to each other, and they stay closer to the observational estimates. Here the OSR increases by 0.5-1 W/m2 at the start of the control-1950 (after small changes in the spinup-1950 period) while the OLR reduces by up to 2 W/m2 (partly in the spinup-1950 and partly in the control-1950 periods) again

consistent with a surface temperature drop of about 0.75 K. The MM model continues to adjust for the first 40 years of control-1950, with reducing ST, before settling into a quasi-equilibrium with global TOA of +0.1-0.2 W/m2 and increasing ST. The MH model only cools very slightly during the control-1950 simulation from its initial conditions, and is the model that has the smallest trends over the control-1950 period and with the smallest deviation globally from the initial conditions in all of the time series. In terms of robustness, the ML and ML models (not shown) broadly follow their ocean resolution equivalents (LL,

MM respectively).

The oscillations in the MM TOA after 100 years are relatively large, and relate to Antarctic sea ice variability and particularly to a large polynya that opens and closes over time in the Weddell Sea. This is a relatively common feature among climate model simulations (Griffies et al. 2015).

The models have not been tuned beyond the shared (common) scientific configuration developed for the HadGEM3-GC3.1

CMIP6 DECK model for a specific long-term TOA radiation, so it is either chance, or some inherent property of this coupled system that enables all models to achieve a relatively small net TOA balance over such a short time. For comparison, the equivalent HadGEM3-GC3.1 DECK pre-industrial control (piControl) simulations have mean TOA of 0.2 and 0.31 W/m2 over several hundred years (LL, MM resolutions after 652 and 353 years of spinup respectively, Menary et al. 2018).



The zonal mean TOA, OSR and OLR biases compared to the Clouds and the Earth's Radiant Energy System Energy Balanced and Filled product surface fluxes edition 4.0 (CERES-EBAF, Kato et al., 2013) are shown in Fig. 3. This confirms that the largest differences occur when the resolution is changed from LL to any other resolution. At low and mid-latitudes the TOA bias is relatively small, but this is due to a compensation between OSR and OLR components, with OLR biased negative at

nearly all latitudes and OSR biased positive. All models have positive OSR biases at mid-latitudes apart from in near Antarctica where there is a negative bias.

The spatial patterns of cloud radiative forcing (CRF) biases against CERES-EBAF (Kato et al., 2013) are shown in Fig. 4 for both shortwave and longwave components (SW, LW). The large-scale patterns of bias are consistent across model resolutions, but there are significant regional differences. As we increase the resolution, for the Atlantic basin there are reductions in the

SW CRF bias in both the tropics and over the Gulf Stream-North Atlantic Current, in the stratocumulus/upwelling regions (off the west coasts of South America, South Africa and North America) and in the western Pacific. In contrast there is an increase in bias in the eastern tropical Indian Ocean and over South America. For the LW CRF there is less change, the bias in the tropical Atlantic changes from a negative bias to the south of the equator to a positive bias to the north, which links to the precipitation biases shown later. A negative bias also increases with increased resolution in the eastern Indian Ocean (perhaps

somewhat compensating the SW CRF bias).

The time series of sea ice area in the Arctic and Antarctic for March and September (approximately the extremes of the respective seasonal cycles) are shown in Fig. 5, together with some observational estimates (HadISST1.2, Rayner et al., 2003; HadISST.2.2.0.0, Titchner and Rayner 2014), noting that the observations are representative of years 1990-2009, while the model is simulating 1950. As noted previously, the LL model has a large increase in Arctic sea ice area over the initial decades

before approaching the climatology of the other models. All models have somewhat more Arctic sea ice in both summer and winter than observations. In the Antarctic winter the MM model displays considerable decadal variability (particularly later in the simulation, as noted in the TOA variability previously) which is not so evident at other resolutions and is due to a large polynya opening and closing in the Weddell Sea. Summer Antarctic sea ice values are improved over previous versions of the model, particularly at MM resolution for which large-scale warming of the Southern Ocean has been a persistent model bias.

This has been achieved primarily from reduced atmospheric flux biases (Bodas-Salcedo et al. 2016; Hyder et al. 2018; Williams et al. 2017), though a warm bias does still remain (see Fig. 7).

The sea ice area and volume seasonal cycles are shown in Fig. 6 at the end of both the spinup-1950 and control-1950 simulations. There are few observational means to assess sea ice volume, and so here we use the Pan-Arctic Ice Ocean Modelling and Assimilation System sea ice reanalysis (PIOMAS; Zhang and Rothrock, 2003; Schweiger et al., 2011) model

as a reference in the Arctic (1990-2009) and satellite estimates from ICESat (Ice, Cloud,and land Elevation Satellite) for the Antarctic during 2003-2008 (Kurtz and Markus, 2012). The seasonal cycle amplitude and phase of sea ice area is well captured in the models except for LL in the Arctic which has too much sea ice. All the models have more sea ice volume than is indicated by the PIOMAS model and ICESat in the Arctic and Antarctic respectively. In the Arctic the volume increases over time in





the MM and MH simulations while reducing somewhat in LL, while in the Antarctic the MM volume starts lower than the other models but adjusts to a similar mean state.

## 3.2 SST adjustment and biases

The SST biases at the end of the control-1950 period (averaged over years 50-100) are shown in Fig. 7 for each model

resolution. These are shown both as model bias compared to initial condition (top row), and as inter-resolution differences, for the total change (due to atmosphere+ocean resolution), change due to atmosphere resolution alone, and change due to ocean resolution alone (rows two, three, four respectively). A common feature across the models is a generally cold mid-latitude bias, which may partly reflect the experimental design of using EN4 1950-54 initial conditions, the short spinup-1950 period and constant 1950's forcing derived from CMIP6, but is also a feature found in many climate models and experiments (see

Flato et al. 2013, Fig. 9.13; Kuhlbrodt et al. 2018).

The biases typical for a 100 km ocean model (Danabasoglu et al. 2014) are evident in Fig 7(a) for LL, and also described in Kuhlbrodt et al. (2018). They are strongest over the boundary currents in the North Atlantic and North West Pacific, with cold biases of more than 5K, and over the tropics with a cold bias of 1-2K. Warm biases over the stratocumulus decks to the west of Southern Africa, South America and California are also evident. Comparing this to the 25 km model (MM, Fig. 7(b,d)),

there are large reductions in both boundary current and tropical biases, and some reductions in the warm stratocumulus biases. These come at the expense of an enhanced warm bias in the Southern Ocean, which is also common in models at this resolution (Bodas-Salcedo et al. 2014; Flato et al. 2013, Fig. 9.2b), and is due to errors in both the atmospheric fluxes (Bodas-Salcedo et al. 2014; Hyder et al. 2018), and due to the intermediate regime in the Southern Ocean in which the ocean models have no eddy parameterisation but also a poor explicit representation of eddies (Hallberg et al. 2013; Ashby et al. 2019). Although still

sizable, the bias has been considerably reduced compared to previous versions of the model (Williams et al. 2017).

Comparing the 8 km and 25 km ocean models (HH, MM, Fig. 7(e)), there is further warming in the Atlantic in the eddy-rich model with mixed results compared to the bias, and some further cooling in the stratocumulus regions. The Southern Ocean warm bias is slightly reduced, but remains larger than the LL bias.

There is no clean way to attribute the biases to either atmosphere or ocean resolution due to complex coupled interactions, but

rows 3-4 of Fig. 7 shows the impact of atmosphere and ocean resolution change only, respectively, for a given resolution of the other component. The largest changes are found between the L and M resolution components with smaller changes at higher resolutions (consistent with Roberts CD et al. 2018). For the L and M resolution ocean, a higher resolution atmosphere tends to produce a cooler ocean SST, particularly in the ocean upwelling regions, the Southern Ocean and the North Atlantic. A higher ocean resolution, meanwhile, tends to produce a warmer SST particularly in boundary currents and other high gradient

regions, with changes over 6 K. Differences between HH and MH, HM are relatively smaller, though the improved separation of the Gulf Stream from the North American coast is evident in the dipole of SST change with the eddy-rich ocean (Fig. 7k).





To contrast the biases above to model surface drift, and hence the effectiveness of the short spinup experimental design, Fig. 8 shows the SST differences between the start and end of the control-1950 simulation. The different model resolutions evolve in distinct ways. LL, ML (Fig. 8a,b) both warm in the northern Pacific and Atlantic basins, the latter consistent with the reduced Arctic sea ice as seen earlier, and an increasing AMOC and northward heat transport (see later). MM, HM (Fig. 8c,d) both
cool at high latitudes in the Southern Hemisphere, and warm slightly in the northern North Atlantic - the HM model has some additional cooling in the south-east Pacific, likely associated with reduced coastal warm bias. MH, HH (Fig. 8e,f) both have a slight warming in the South Atlantic near the Antarctic Circumpolar Current. However it is clear that the magnitude of this drift is considerably smaller than the mean bias shown in Fig. 7, such that the bias plot at the end of spinup-1950 (not shown) is little different from Fig. 7. The only notable region with similar magnitude of drift and bias is the LL North Atlantic warming
and part of the Southern Ocean cooling in MM and HM.

The annual mean 2 m temperature biases over land are shown in Fig. 9, using model means over years 50-100,  compared to the Climate Research Unit series 4.01 data set for period 1940-1960 (CRU TS; Harris et al., 2014). The warmer SSTs and reduced sea ice extent in the Arctic Ocean between LL and the higher resolution models give significantly warmer surface temperatures over Scandinavia, northern Russia and Alaska of 2-3 K. Tropical cold biases are also reduced as resolution is
increased, leading to the HH model having a considerably smaller global root mean square (RMS) bias.

The results above indicate that, for the surface climatology, the HighResMIP simulations are adequately long to illustrate robust differences due to model resolution. However, the deeper ocean requires much longer to come into any pseudo-equilibrium, and these drifts are described next.

### 3.3 Deep ocean evolution

The evolution of subsurface ocean changes from the initial state (henceforth referred to as ocean drift) at different model resolutions are shown in Figures 10 and 11 for temperature and salinity respectively, for both the global ocean (left column) and the Atlantic basin (40°S - 70°N excluding the Mediterranean, right column), for all model resolutions with a spinup-1950 simulation. For the global ocean, the temperature and salinity drifts differ mainly in their magnitude, while the Atlantic drifts have different patterns. These drifts may be compared to those in Small et al. (2014), Griffies et al. (2015) and Kuhlbrodt et
al. (2018).

All models have a global cooling of 0.5-1 K over the top 200 m (Fig. 10 left column) which gradually reduces over time, and a warming centred at 800 m, with an enhanced magnitude in the L ocean models. The LM, MM models have a deep warming not seen in other models, which is likely associated with the Southern Ocean bias (not shown). The global salinity drift (Fig. 11 left column) shows all models have a freshening over the top 300 m with magnitude increasing over time. They also have
an increase in salinity between 1000-2000 m, which is larger with the L ocean.

The temperature evolution in the Atlantic (Fig. 10 right column) is similar to the global, with near surface cooling and a warming at mid-depths, both of which are stronger in the L ocean. The warming at mid-depths is also associated with an increase in salinity (Fig. 11 right column) which is also seen in Griffies et al. (2015) and Kuhlbrodt et al. (2018), and is likely

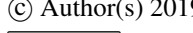



to be associated with the AMOC circulation, production of North Atlantic Deep Water (NADW) and biases in representing deep overflows in the North Atlantic (Danabasoglu et al. 2014). The change from L to a higher resolution ocean causes the surface Atlantic salinity bias to switch from a freshening to a positive increase, with the eddy rich MH resolution notable for having greatly reduced salinity drifts in the Atlantic - these differences are consistent with differences in (tropical) precipitation

as shown later.

Figs. 10c,d, and 10e,f illustrate the impact of atmosphere and ocean resolution on these drifts. Using a higher atmosphere in ML (Fig. 10d) suggests a reduction in the Atlantic mid-depth warming (perhaps an improvement in the NADW), while LM (Fig. 10e,f) shows a slight reduction in the magnitude of warming at 1000 m, and the warming of the bottom waters globally associated with the Southern Ocean bias.

The above has shown that 100 years is not sufficient to saturate the deep ocean drifts, as would be expected. However, some differences with resolution do seem to be robust and possibly linked to process improvement, particularly in the Southern Ocean where the eddy-rich MH simulation greatly reduces the deep warm temperature drift, and in the North Atlantic for both temperature and salinity.

The eddy-rich ocean simulation, MH, has both the minimum in surface adjustment (of radiation and temperature), as well as

having the smallest deep ocean drifts from the EN4 initial conditions. It is unclear whether this is due only to improved representation of key processes (for example in the Southern Ocean and Atlantic), or whether by chance it is better able to adjust from these initial conditions. The improvements show some similarity with those in Griffies et al. (2015), and hence we need a multi-model study using the HighResMIP experimental design to firmly establish whether such an increase in ocean resolution can robustly reduce deep ocean drifts, and hence establish whether higher resolution (ocean/coupled) models require

less spinup time - this work is ongoing as part of PRIMAVERA.

### 3.4 Mean state precipitation biases

Changes in the annual mean precipitation bias against GPCP2.3 1979-2014 (Adler et al. 2018) with resolution are shown in Fig. 12, averaged over model years 50-100 of control-1950, and are generally consistent with those found in the multi-model analysis of Vanniere et al. (2018). The mean biases are shown in the top row, together with the total differences

(atmosphere+ocean resolution changes, row two), and then the impact of atmosphere and ocean resolution changes individually (rows three and four respectively).

Some aspects of the large-scale biases are common across all resolutions consistent with those shown in Williams et al. (2017). There is excessive precipitation in the western North Pacific, south-east Asia, the western Indian Ocean, and the South Pacific Convergence Zone (SPCZ), which also includes a double ITCZ error in the east Pacific (Lin 2007).

In the tropical Atlantic, there is a dipole error of 2-3 mm/day across the equator in LL with too much precipitation in the south and too little in the north, with significant consequences for land precipitation over South America. This error is markedly reduced in the MM model and further reduced to the south of the equator in HH, together with a reduction in the dry bias over West Africa. In the Pacific the LL model has a double ITCZ error which reaches to the eastern Pacific boundary, which is





reduced at higher resolutions at the expense of a further increases in excessive precipitation both to the north and south of the equator. There is a reduction in the dry bias at the tropical west Pacific at higher resolutions and some improvement over the Maritime Continent and India, with an increased dry bias in the eastern Indian Ocean. The subpolar North Atlantic has increased precipitation at higher resolutions, primarily due to the ocean and likely associated with a warmer SST.

Examining the respective roles of atmosphere and ocean resolution (Fig. 12(f,g,h) and (i,j,k) respectively), a higher resolution atmosphere tends to be dry to the south of the equator and wet to the north with a similar magnitude at each ocean resolution, while the ocean resolution change is much bigger between L and M.

The change in tropical Atlantic precipitation is consistent with an ITCZ located further north at higher resolution. This would be consistent with increased AMOC at higher resolution causing SST gradient changes (Jackson et al. 2015, also Fig. 7(d,e)),

and may also be a consequence of a change in the global energy budget leading to shifts in the Hadley Cell/ITCZ position (Bischoff and Schneider 2016). The precipitation and consequent evaporation changes in the tropical Atlantic are also consistent with the salinity drifts shown in Fig. 11.

### 3.5 Atlantic Ocean meridional circulation and transports

The Atlantic Northward Heat Transport (NHT) and Atlantic Meridional Overturning Circulation (AMOC) time series are

shown in Figures 13 at 26.5°N, and are calculated in a consistent way to observations at the RAPID-MOCHA array 2004-2017 (Smeed et al. 2017; Johns et al. 2011) using the RapidMoc algorithm described in Roberts et al. (2013) and Roberts (2017). The MM and MH models increase both AMOC and NHT over the 30 year spinup-1950 period, and subsequently vary about this mean state with no significant further drift evident. The LL model has decreasing transport over the spinup-1950 period, but over the 100 year control-1950 period both AMOC and NHT gradually strengthen to a state which is then maintained

stably for many 100's of years. This state is such that the AMOC strength is similar to MM, but the NHT remains about 10% lower. This evolution is consistent with the SST drift discussed above, with increasing AMOC and NHT gradually warming the North Atlantic.

The MH and HH models have NHT which is most consistent with the observations, suggesting an important role for an eddy-rich ocean improving boundary current representation (Treguier et al. 2012; Roberts et al. 2016). However, for AMOC

transports, the observations are perhaps more consistent with the lower resolution models. This apparent inconsistency has been investigated previously by Msadek et al. (2013) and Roberts CD et al. (2018). Using a breakdown of heat transport components from the RAPID-MOCHA array (Smeed et al. 2017; Johns et al. 2011), they showed that the amount of NHT was typically underestimated due to both a too-weak AMOC, and too little heat transport per Sverdrup of AMOC strength. The same breakdown for the model resolutions is shown in Fig. 14 as well as the AMOC profile with depth. All the models have a

weaker overturning component to the heat transport than indicated by RAPID-MOCHA, though the H ocean does shift to higher values than the lower resolutions. Other models and reanalyses show a similar relationship, with strong correlations between AMOC and NHT, producing a regression that would imply, for the observed AMOC strength, a weaker NHT than observed (Danabasoglu et al. 2014; Roberts et al. 2013; Msadek et al. 2013).





The depth profile of AMOC indicates a strengthening at most depths with increased ocean resolution, with all models having a maximum at around 1000 m consistent with the observations, and the higher resolution models better agreeing with observations at depth but becoming too strong at 1000 m. The seasonal cycle of AMOC with annual mean removed (Fig. 14) indicates the higher resolution models can match the observed magnitude. The AMOC minimum in March corresponds to the

period of maximum variance, with reduced variability in summer.

The NHT dependence on latitude in the North Atlantic is shown in Fig. 15, where the individual components (total, time-mean, time-varying and eddy-induced) are also indicated. For the LL model, the time-varying component is only visible near the equator, while the eddy-induced transport associated with the Gent-McWilliams parameterisation reaches ~0.1 PW around 40°N. The MM and HH models have similar time-varying components to each other, but the eddy-rich ocean has considerably

stronger mean transport which better agrees with observations between the equator and 40°N. One aspect of note is the increased HH NHT northwards of 45°N towards the Arctic as also seen in Roberts et al. (2016). It is unclear if this is excessive compared to observations, but if so it would imply that the ocean does not lose enough heat to the atmosphere at these latitudes.

### 3.6 Antarctic Circumpolar Current

The time evolution of the Antarctic Circumpolar Current (ACC) transport, calculated as the volume transport through Drake

Passage, is shown in Fig. 16. The mean net eastward transports of 155, 90 and 125 Sv respectively for LL, MM and MH models compares to the recent observational range of 173±11 Sv (Donohue et al. 2016), with earlier estimates lacking a robust barotropic component (e.g. 137±8 Sv; Cunningham et al. 2003). Using the former measure, LL is closest to the observational range, while MM is only 40% of it. A part of the deficit in the M ocean model is due to a strong counter-current around the Antarctic shelf of about 20 Sv, together with changes to the density front, as discussed more fully in Menary et al. (2018).

Figure 16 indicates that the impact of different atmosphere resolutions is small compared to the impact of ocean resolution. Despite a reduced transport, however, the frontal structures associated with the ACC, some with a barotropic structure, are much more evident in the M and H ocean models (not shown).

### 3.7 ENSO variability

As the dominant mode of interannual tropical variability, El Niño-Southern Oscillation (ENSO) is a key aspect of climate

variability with worldwide impacts (Timmermann et al. 2018). Over time there has been some improvement in modelling ENSO in global climate models (e.g. Bellenger et al. 2014), with HadGEM3-GC3.1 performance described in Williams et al. (2017), Kuhlbrodt et al. (2018) and Menary et al. (2018). Fig. 17 shows the power spectrum of Niño3.4 monthly surface temperature anomalies, calculated using a periodogram method with 50 years of data in each sample, and a 25 year overlap between samples, with the average power spectrum and range (shading) shown. It is clear that 50-100 years of sampling is not

enough to give a robust measure of the power (as indicated by the large ranges in LL and MM simulations), but for the simulations with 150+ years, the mean spectrum agrees well with that from HadISST1.1 observations for 1877-2018 (Rayner



et al. 2003). The standard deviations of the mean Nino3.4 DJF SST from the models are all slightly higher than the observed value of 0.93.

The composite December-January-February (DJF) mean surface temperature patterns relating to El Niño and La Niña events are shown in Fig. 18, with events defined when the DJF Niño3.4 index exceeds ±0.7 K. There is a robust pattern to the global

surface temperature anomalies which agree well (over the ocean) with the observed HadISST1.1 dataset and over the land with the Climate Research Unit time series 4.01 data set (CRU TS; Harris et al., 2014) 2 m temperatures for 1901-2016 (Fig. 18f). The extension of the El Niño and La Niña patterns past the dateline is slightly excessive in the LL model, which is a common bias (e.g. Guilyardi, 2006; Roberts CD et al., 2018). The teleconnections to land surface temperature anomalies are robust over the Americas and Africa, but less so over Eurasia; the models with H atmosphere tend to have stronger negative anomalies

over Northern Europe with El Niño, but these time series are shorter and hence have far fewer events.

The equivalent composite rainfall patterns are shown in Fig. 19 for the models and GPCP2.3 observations for 1979-2014 (Adler et al. 2018) for El Niño and La Niña events. The extension of the SST pattern into the western Pacific in LL is also evident here as excessive precipitation at the equator in the West Pacific, with some improvement at higher resolutions. All model resolutions mirror the observed teleconnections quite faithfully, though the dry anomaly with El Niño events over South

Africa is not robustly captured.

The near 1:1 ratio of El Niño to La Niña events found in observations is replicated in the models, but the ratio of Cold Tongue (CT, East Pacific) to Warm Pool (WP, Central Pacific) events, as defined by the indices in Ren and Jin 2011, is less well represented, as noted in Fig. 18 titles. The LL model has a near 1:1 ratio of such events compared to the observed 2:1, the higher resolution ocean models have far fewer WP events compared to CT but there seems to be little systematic change with

resolution.

There seems to be only minor differences in the ENSO performance in the models at different resolutions, mainly in slight differences to the SST composite. It is clear that 100 years is not long enough to assess the power spectrum, as noted previously by Wittenberg (2009) and Stephenson et al (2010), but the composite patterns of surface temperature and precipitation show relative robustness.

**4 Summary and discussion**

As part of the CMIP6 HighResMIP project, a wide range of coupled model simulations with atmosphere resolutions between 250 km and 50 km, and ocean resolutions from 100km to 8km, have been performed with the HadGEM3-GC3.1 model. We have shown that increased model resolution in the atmosphere and ocean can have considerable impact on climate model biases of the mean state and variability, both at the surface in terms of temperature and precipitation, as well as in the deeper ocean.

We have demonstrated that the new CMIP6 HighResMIP experimental design, with only a multi-decadal spinup and 100 year simulation length, is sufficiently long to robustly establish some of these responses in model bias. This has enabled the use an eddy-rich 8 km ocean model within the same suite of experiments, to make a more comprehensive chain of resolutions, and





hence further test the robustness of our results. These experiments may also enable better understanding of the model adjustment process (so-called spinup from initial conditions), which tends not to be a focus of the standard CMIP simulations with a long pre-industrial spinup. This makes it harder to understand why the deep ocean adjustment process timescales may be different with different resolutions, and what role these biases might play in model sensitivity to changes in forcing.

We find that increased ocean resolution is key to reducing many of the most common SST biases, while a combination of ocean and atmosphere resolution significantly improves the large tropical Atlantic precipitation biases seen in typical CMIP-resolution models, the latter having the potential to cause considerable uncertainty in projections of future rainfall changes.

We have also found some potential links between the biases and the evolution of the mean state. Based on previous work, it seems likely that the strengthened Atlantic Meridional Overturning Circulation (AMOC) and northward heat transport in the

tropical Atlantic is linked to the improved SST biases and reduced precipitation (and ITCZ) biases, which in turn may be associated with some of the deeper ocean biases. These may also link to the different spinup behaviours seen in the different models. The initial drop in AMOC and Northward Heat Transport in the LL model causes a cooling in the North Atlantic and Arctic, with a consequent increase in sea ice. Over time the stronger temperature (and salinity) contrast between equator and pole drives an increase in AMOC which gradually warms the Arctic. This increase in AMOC could be enhanced by the

increased tropical Atlantic salinity bias in LL, which would increase the density of water reaching the northern North Atlantic and enable a stronger AMOC circulation to develop. Using the same initial conditions and short spinup in all experiments may enable better understanding of such adjustment processes than is generally possible in standard CMIP simulations.

While representing a substantial improvement over the length of simulation period typically used in global high resolution experiments in the past, there are aspects of these simulation for which 100 years is not sufficient. The LL model in particular

seems to take considerably longer than 30 years to quasi-equilibrate aspects of the large-scale circulation (such as AMOC), perhaps indicating that some processes are inadequately represented at this resolution.  The deep ocean equilibration timescales are clearly much longer than 100 years, although there is evidence that the different resolution models trending towards different final states and the magnitude of the drifts is resolution dependent. This 100 year period is still not enough for a robust estimate of the ENSO power spectrum and variability, although the ENSO composites of surface temperature and

precipitation teleconnections are apparently robust. The longer control-1950 simulations (at the lower resolutions) have been vital to test the reliability of these assertions, and hence there is probably a role for such longer simulations within the HighResMIP experimental design. Such timescales also emphasise the importance of considering both the control-1950 and hist-1950 simulations when evaluating the models involved in HighResMIP, to properly assess model drift compared to response to forcing.

It is unclear how robust all of the results shown here will be across a multi-model dataset. Ongoing work within Horizon 2020 PRIMAVERA project (Grist et al. 2018; Roberts CD et al. 2018; Vanniere et al. 2018) suggests that at least for changes from 100 km to 25 km ocean resolution there are robust reductions in SST and precipitation biases, while work by Griffies et al. (2015) and Small et al. (2014) do indicate some consistency in further changes to eddy-rich ocean resolutions. Further work using this multi-model ensemble within HighResMIP, in addition to comparing these simulations to their CMIP6 DECK

equivalents, in ongoing and may reveal further insights into the impact of resolution and model complexity (Earth System processes such as interactive aerosols, biogeochemistry) and perhaps indicate the best trade-offs for gaining the largest model improvements for the smallest computational cost.

**Data and code availability**

Most of the model data used in the following analysis is available from the CMIP6 Earth System Grid Federation, and can be located using the information in Roberts M (2018b, 2017a, 2017b, 2017c, 2017d) for resolutions LL, MM, HM, MH, HH respectively. Other model resolutions are not part of the official HadGEM3-GC3.1 CMIP6 HighResMIP submission, and this data is available on request via the CEDA-JASMIN platform.

Due to intellectual property rights restrictions, we cannot provide either the source code or documentation papers for the UM
or JULES. The MetUM is available for use under licence. For further information on how to apply for a licence please contact um_collaboration@metoffice.gov.uk. JULES is available under licence free of charge. For further information on how to gain permission to use JULES for research purposes see https://jules.jchmr.org/software-and-documentation. The model code for NEMO v3.6 is available from the NEMO website (http://www.nemo-ocean.eu). On registering, individuals can access the code using the open-source subversion software (http://subversion.apache.org/). The model code for CICE is freely available
(http://oceans11.
lanl.gov/trac/CICE/wiki/SourceCode) from the United States Los Alamos National Laboratory. In order to implement the scientific configuration of GC3.1 and to allow the components to work together, a number of branches (code changes) are applied to the above codes. Please contact the authors for more information on these branches and how to obtain them.

**Author contributions**

Model simulations were performed by MJR, AC, RS. JS post-processed the model output into CMOR format for publication to ESGF. MJR prepared the manuscript with contributions from all co-authors.

**Competing interests**

The authors declare that they have no conflict of interest.

**Acknowledgements**

MR, JS, PLV, RS, BV, AB, CDR acknowledge PRIMAVERA funding received from the European Commission under Grant Agreement 641727 of the Horizon 2020 research programme. MR, HH, EB, PM, LJ were supported by the Met Office Hadley Centre Climate Programme funded by BEIS and Defra (GA01101). TK was funded by the National Environmental Research Council (NERC) national capability grant for the UK Earth System Modelling project, grant NE/N017951/1; and by the EU Horizon 2020 Research Programme CRESCENDO project, grant agreement 641816. PLV, RS, AB, BV acknowledge funding



from the National Environmental Research Council (NERC). AC acknowledges funding from the ACSIS project that is supported by the Natural Environment Research Council [grant number NE/N018044/1].

Data from the RAPID-MOCHA program are funded by the U.S. National Science Foundation and UK Natural Environment Research Council and are freely available at www.noc.soton.ac.uk/rapidmoc and

https://mocha.rsmas.miami.edu/mocha/results/mocha-new/index.html.

We acknowledge extensive use of the supercomputers at the Met Office and the ARCHER UK National Supercomputing Service to complete the simulations, and for the CEDA-JASMIN platform to enable data storage and model analysis.



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





**Tables**

| Ocean grid | ORCA1 | | ORCA025 | | ORCA12 | |
|---|---|---|---|---|---|---|
| Ocean timestep (min) | 30 | | 20 | | 5 | |
| Gent-McWilliams; GM coeff; isopycnal diff | On; variable; 1000 | | Off; 0; 150 | | Off; 0; 125 | |
| Momentum dissipation | Laplacian; $20 \times 10^3$ m$^2$s$^{-1}$ | | Bilaplacian; $-1.5 \times 10^{11}$ m$^4$s$^{-1}$ | | Bilaplacian; $-1.25 \times 10^{10}$ m$^4$s$^{-1}$ | |
| Snow on sea ice albedo: near-infrared; visible | 0.68; 0.96 | | 0.7; 0.98 | | 0.7; 0.98 | |
| Atmos timestep (min) | 20 | 15 | 15 | 10 | 15 | 10 |
| USSP launch factor (for QBO) | 1.3 | 1.2 | 1.2 | 1.2 | 1.2 | 1.2 |
| Experiments submitted to CMIP6 ESGF | spinup-1950; control-1950, hist-1950 | spinup-1950 control-1950 | spinup-1950; control-1950, hist-1950 | control-1950, hist-1950 | spinup-1950; control-1950 | control-1950, hist-1950 |
| Atmos model name (mid-latitude grid spacing) | N96 (135 km) | N216 (60 km) | N216 (60 km) | N512 (25 km) | N216 (60 km) | N512 (25 km) |
| CMIP6 nominal resolution (atmosphere, ocean) | 250 km, 100 km | 100 km, 100 km | 100 km, 25 km | 50 km, 25 km | 100 km, 8 km | 50 km, 8 km |
| HadGEM3-GC31 naming convention | **LL** | **ML** | **MM** | **HM** | **MH** | **HH** |

Table 1: Parameter value changes between different model ocean (from top) and atmosphere (from bottom) resolutions, together with resolution naming conventions.





| Model name | CMIP6 resolution (atmos-ocean) km | No. years | Nodes (atmos-ocean) | Max turnaround (years per day) | Output per year (TB) |
|---|---|---|---|---|---|
| LL | 250-100 | 1000 | 12-2 | 4 | 0.13 |
| MM | 100-25 | 650 | 50-24 | 1.3 | 0.73 |
| HM | 50-25 | 100 | 90-24 | 0.5 | 2.8 |
| MH | 100-8 | 160 | 34-171 | 0.45 | 2.0 |
| HH | 50-8 | 100 | 90-171 | 0.4 | 4.5 |

5    Table 2: Costs of various model resolutions on a Cray XC40 with 36 cores per node, together with raw model output volumes.



**Figures**

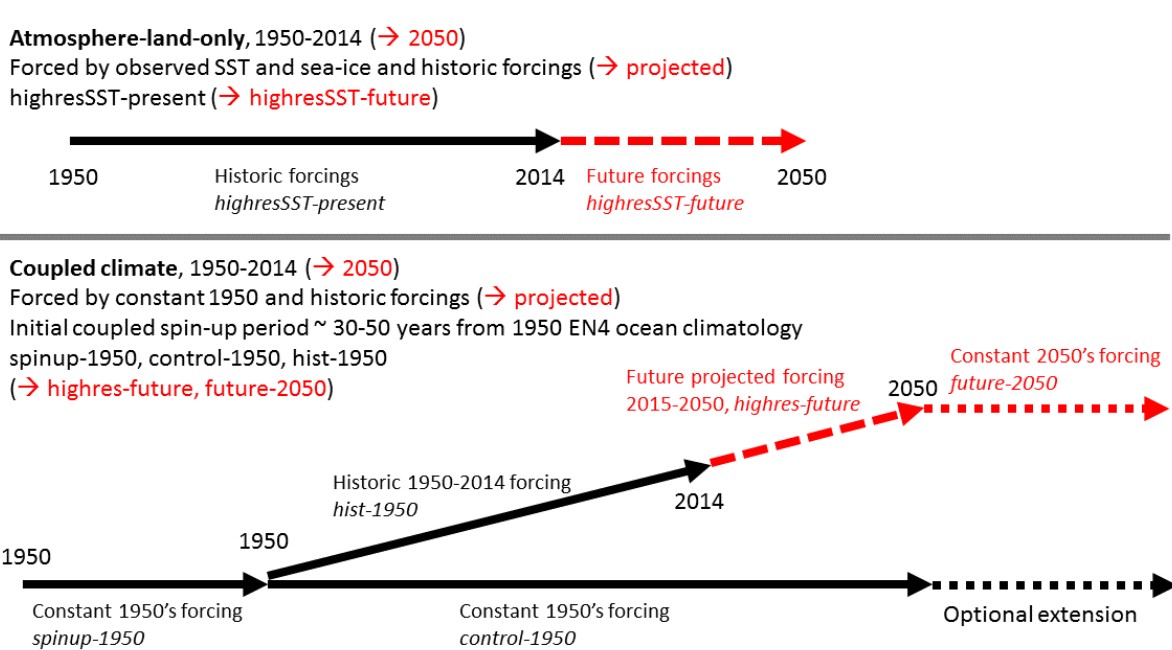

Figure 1: The HighResMIP experimental design, including the names of the component experiments and an indication of their relationship.



Figure 2: Time series of globally averaged quantities for three ocean resolution spinup-1950 simulations (x-axis nominal years 1920-1950 here), and five control-1950 simulations initialised from these conditions (x-axis nominal years 1950-2300). (a) Top of Atmosphere (TOA) radiation, W m$^{-2}$; (b) Outgoing Shortwave Radiation (OSR) , W m$^{-2}$; (c) Outgoing Longwave Radiation (OLR) , W m$^{-2}$; (d) surface temperature, K. The horizontal black lines are estimates (solid) and uncertainty (dashed) of fluxes from Wild et al. (2013).

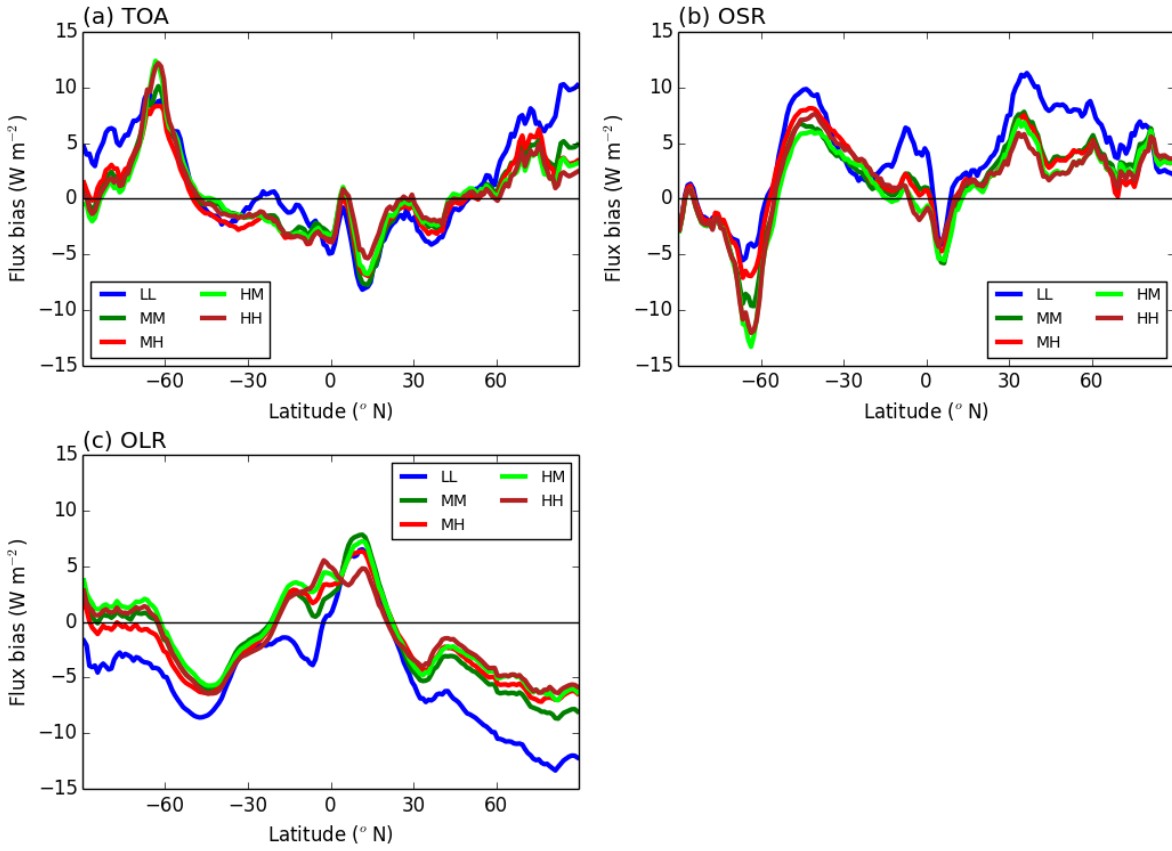

Figure 3: Annual mean zonal mean model radiation biases (years 50-100) of Top of Atmosphere radiation components against CERES-EBAF observations 2000-2018 (Kato et al. 2013). (a) Top of Atmosphere net radiation; (b) Outgoing shortwave radiation; (c) Outgoing longwave radiation.

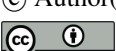





Fig. 4: Annual mean model bias over years 50-100 in cloud radiative forcing (Wm$^{-2}$) for (left column) shortwave cloud forcing and (right column) longwave cloud forcing, compared to CERES-EBAF observations 2000-2018 (Kato et al., 2013).

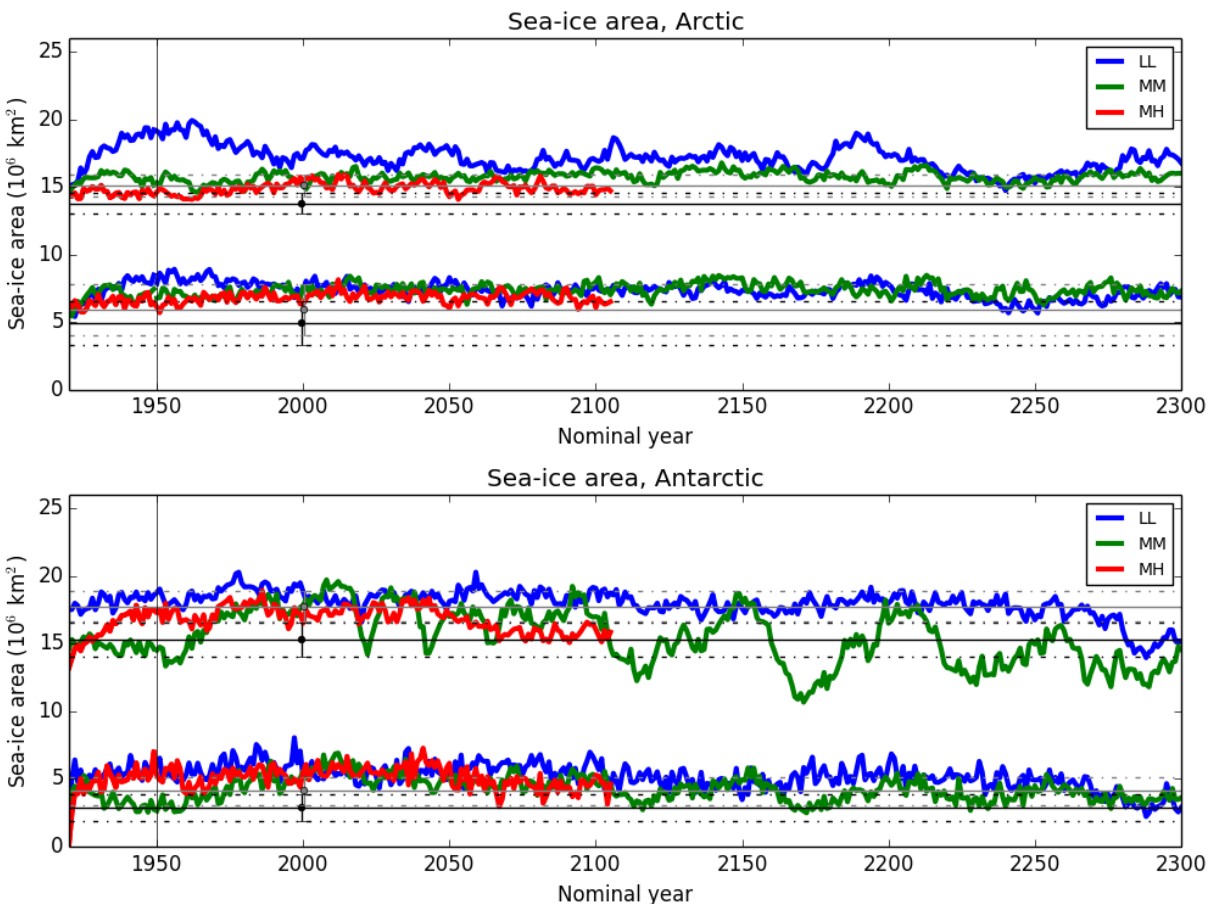

Figure 5: Time series of sea ice areas in the (top) Northern and (bottom) Southern Hemispheres for three model resolutions, for spinup-1950 and control-1950 simulations. In each hemisphere, the winter and summer months (March, September in NH, September, March in SH) are shown by the upper and lower groups of contours respectively. Observations are HadISST1.2 (black line) and HadISST.2.2 (gray line), with their mean value over 1990-2009 shown as the solid line, and the maximum and minimum during that period as dashed lines.





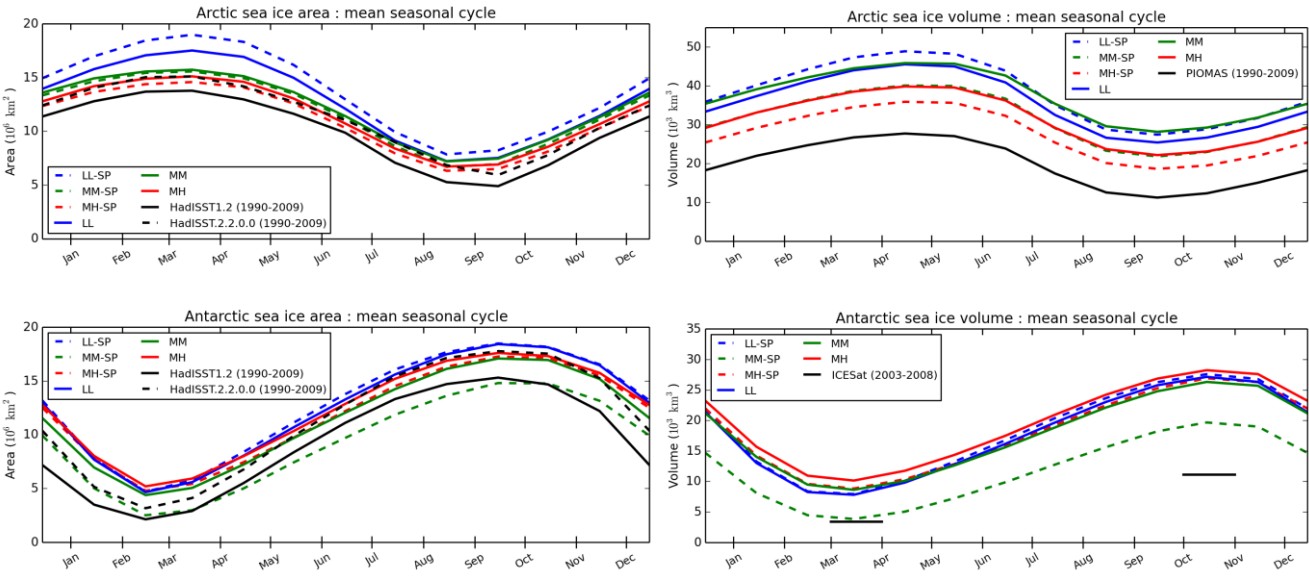

Fig. 6: Seasonal cycles for (left) sea ice area and (right) sea ice volume from models and observations, in the (top) Arctic and (bottom) Antarctic regions. The dashed and solid colours refer to different model periods (dashed is 10 year mean at end of spinup-1950, solid is 20 year mean at end control-1950). Observed sea ice area from HadISST1.2 (1990-2009) and HadISST2.2.0.0 (1990-2009), and sea ice volume from the PIOMAS model (1990-2009) and ICESat (2003-2008).





Figure 7: (top row) Model SST bias in control-1950 years 50-100 vs EN4 (1950-1954 mean); (second row) The total SST differences between MM-LL and HH-MM resolutions; (third row) The impact of changing the atmosphere resolution at each reference ocean resolution; (bottom row) The impact of changing the ocean resolution at each reference atmosphere resolution.

5   Units are K. Points that contain annual mean sea ice are masked.





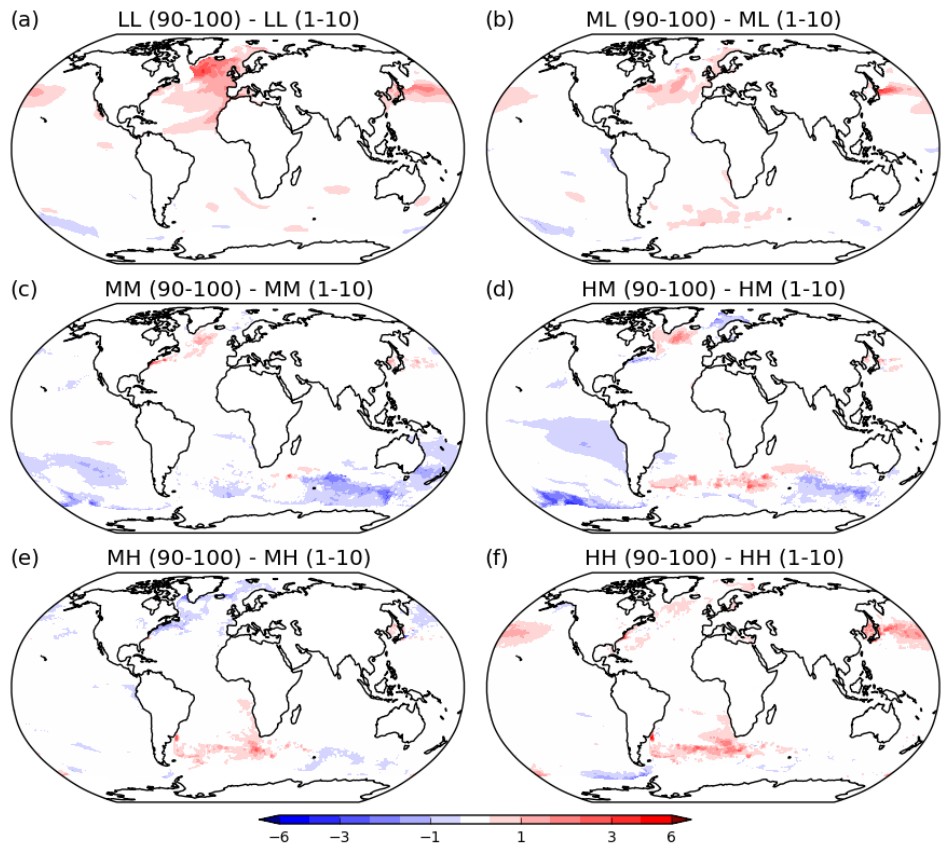

Figure 8: Change in SST over 100 years. (left column) the difference between years 90-100 of control-1950 vs the end of spinup-1950; (right column) the difference between the year 90-100 of HM - MM_spinup, and HH - MH_spinup, that is for the resolutions which did not have their own spinup.





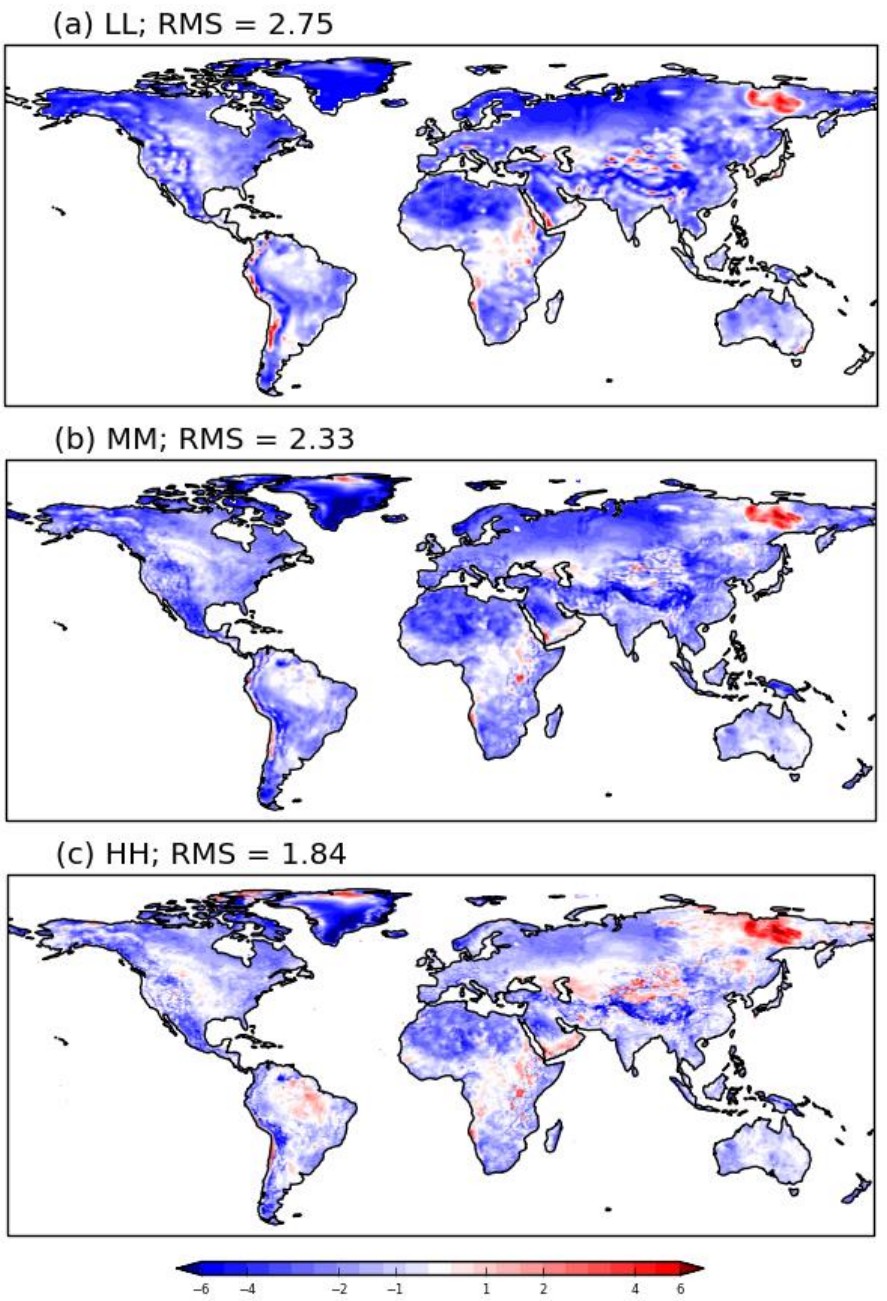

Figure 9: Annual mean bias in 2 m temperature over land (℃) for control-1950 simulations (years 50-100) relative to the Climate Research Unit time series 4.01 data set (CRU TS; Harris et al., 2014) for the period 1940–1960.

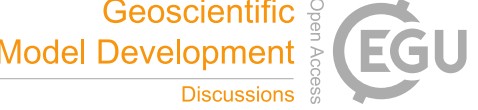

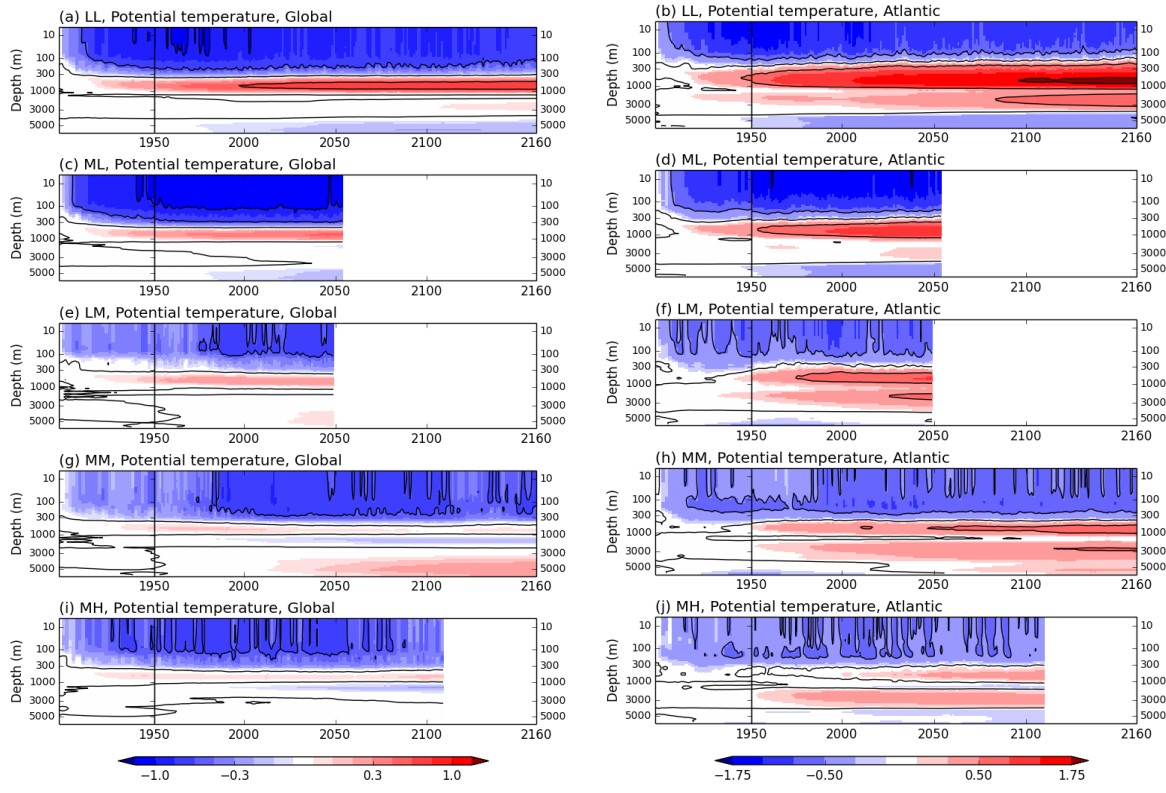

Figure 10: The mean ocean potential temperature anomaly (compared to year 1) on model levels against time, for (left) global and (right) Atlantic ocean basin, from the spinup-1950 period through the first 210 years (where available) of control-1950.

5   Note the different scales for the different regions.



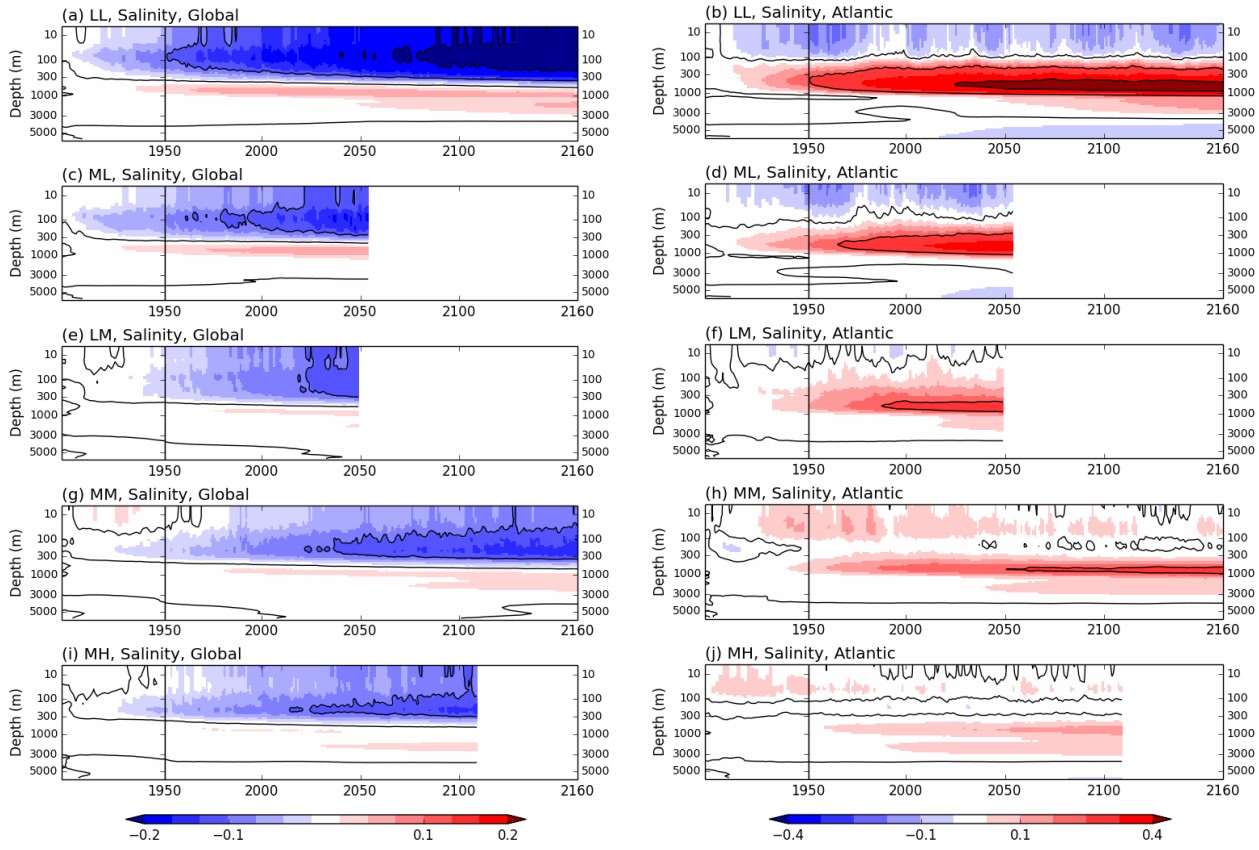

Figure 11: As Fig. 10 but for ocean salinity, again with different scales on left and right.





Figure 12: (top row) Annual mean model precipitation bias of control-1950 model years 50-100 vs GCPC_v2.3 1979-2014; (second row) The total precipitation differences between MM-LL and HH-MM resolutions; (third row) The impact of changing the atmosphere resolution at each reference ocean resolution; (bottom row) The impact of changing the ocean resolution at each reference atmosphere resolution. Units are mm day-1.

Figure 13: (top) Time series of annual mean (January-December) Atlantic northward ocean heat transport at 26.5°N calculated consistent with the RAPID-MOCHA array using the RapidMoc code (Roberts CD 2017). (bottom) As top but for volume transport, i.e. the Atlantic Meridional Overturning Circulation at 26.5°N and 1000 m depth, with net transport across section subtracted. Different model resolutions over the spinup-1950 and control-1950 simulations are shown together with the RAPID-MOCHA observations in black. The thick line shows a 5-year running mean of the annual values (thinner lines). A box and whiskers plot on the right (using the control-1950 data) shows the lower to upper quartile range as the coloured box, the median (black line within that box) and the whiskers show the range of the data (excluding flier points shown by +).





Figure 14: AMOC-related metrics calculated using RapidMoc code (Roberts CD 2017) consistent with the RAPID-MOCHA observations, using model years 1-100. (a, top left) Scatter plot of each annual mean AMOC transport (x-axis), with associated total heat transports (y-axis) decomposed into total (circles), overturning circulation (x) and gyre transport (triangles). (b, top right) Profile of AMOC transport with depth, shading one standard deviation either side of the mean. (c, bottom) The seasonal cycle of AMOC at 26.5°N with annual mean value subtracted. The standard deviation is indicated for March and July only.





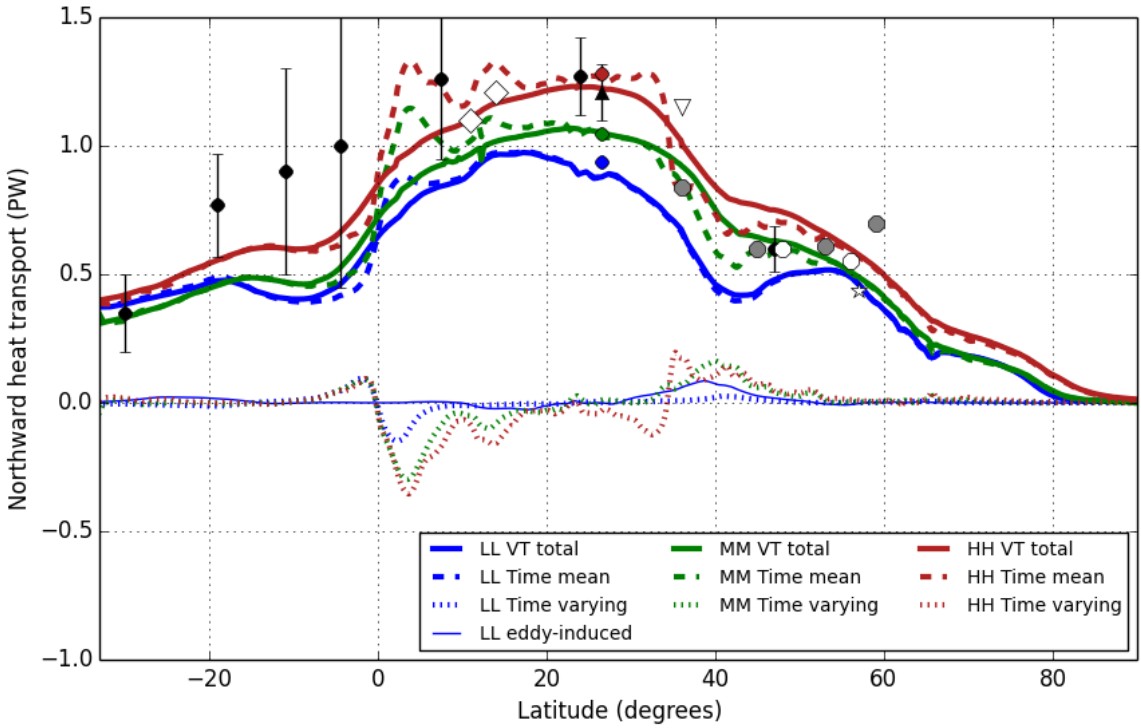

Figure 15: Ocean northward heat transport in the Atlantic, averaged over years 50-100, separated into components: (solid lines); total; (dashed lines): time mean; (dotted lines): time-varying; (thin solid lines): bolus transport. The MM and HH models do not have a separately diagnosed eddy-induced transport as its parameterisation (Gent-McWilliams scheme) is switched off. The black circles and lines are observational estimates from Ganachaud and Wunsch (2003) with uncertainty, the black triangle is RAPID-MOCHA (Johns et al. 2011). Other observations in white and gray symbols: white diamond - Bryden and Imawaki (2001); gray circle - Talley (2003); white open circle - Lumpkin and Speer (2007); white inverted triangle - McDonagh et al. (2010); white star - Lozier et al. (2019). The coloured circles at 26.5°N correspond to the mean values using the RapidMoc calculation as in Fig. 10.



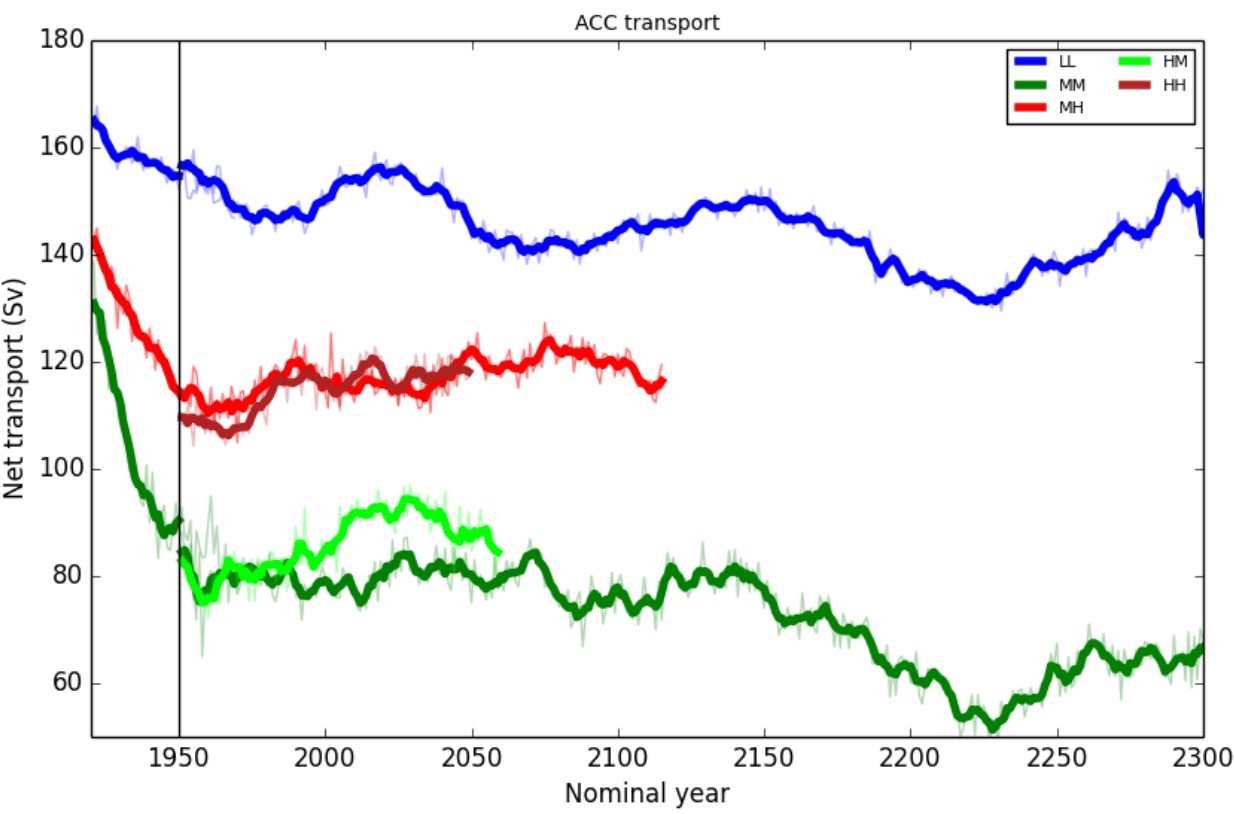

Figure 16: Time series of the Antarctic Circumpolar Current (ACC) transport calculated between Antarctica and South
5    America across Drake Passage for spinup-1950 and control-1950 simulations.



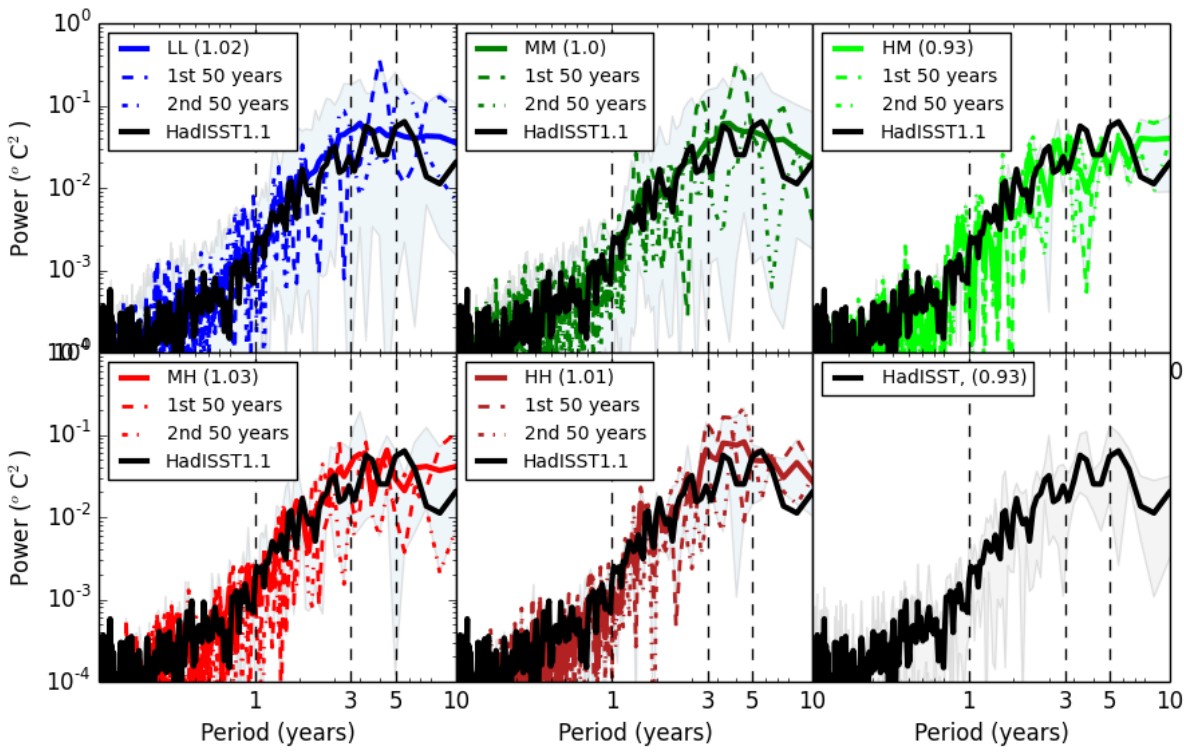

Figure 17: Power spectrum of Niño3.4 surface monthly temperature variability for different resolution models and HadISST1.1 observations. The coloured line shows the mean power spectrum over all years, the dashed line shows the first 50 years, while the shading shows the range over all 50 year sub-sampled periods. The number in brackets is the standard deviation of the DJF mean SST in the Niño3.4 region.







Figure 18: Composite DJF mean surface temperature for (top) El Niño and (bottom) La Niña from models and observations based on El Niño and La Niña events. The number of events sampled is shown in the title, and the proportion of El Niño events classified as Cold Tongue (CT) and warm pool (WP). Events are defined as years when the Niño3.4 DJF SST anomaly exceeds 0.7 K. The observations are a combination of HadISST1.1 over the ocean and CRU 2 m temperatures over land, with values masked over HadISST1.1 sea ice regions.





Figure 19: As Fig. 18 but for DJF composites of mean precipitation (top) El Niño and (bottom) La Niña from models and
GPCP2.3 precipitation. The number of events sampled is shown in the title, and the proportion of El Niño events classified as
Cold Tongue (CT) and warm pool (WP). Events are defined as years when the Niño3.4 DJF SST anomaly exceeds 0.7 K.