# Peer review of "Description of the resolution hierarchy of the global coupled HadGEM3-GC3.1 model as used in CMIP6 HighResMIP experiments"

_Geoscientific Model Development, 2019_

## Referee Comment (RC1) · Stephen M. Griffies (Referee) · 15 Jul 2019

I found this manuscript to offer a readable and well organized description for a variety of simulation features from the Hadley Centre HighResMIP hierarchy of AOGCMs. As a purely descriptive manuscript, I have little to critique since any such effort is subjective. I generally found the authors touched upon most of the key features that climate modelers might find of relevance.

My main concern is with the following statement on page 15, lines 9-10:

"Due to intellectual property rights restrictions, we cannot provide either the source code or documentation papers for the UM 10 or JULES."

That is a very unsatisfying situation that, in effect, means these simulations are not reproducible by any willing and able person outside of the Hadley Centre. I am disappointed that the Hadley Centre has failed to fully embrace scientific programming of the 21st century where open development / open access greatly supports the climate science endeavour by enabling reproducibility.

I recommend publication with the following minor points.

page 6 lines 14 and 15: please give units

page 6: line 28: please add following citation along with Griffies et al 2015:

Preconditioning of the Weddell Sea polynya by the ocean mesoscale and dense water overflows, 2017: C.O. Dufour, A.K. Morrison, S.M. Griffies, I. Frenger, H.M. Zanowski, M. Winton, Journal of Climate, vol. 30, 7719-7737, doi:10.1175/JCLI-D-16-0586.1

page 10, paragraph with line 15: Is worth noting that interior biases can be exacerbated by spurious mixing, especially at the 1/4 ocean resolution. Citations to work from Alex Megan as well as Griffies, Illicak, others can be made.

Figures 7,8,12 could be more useful with statistical information such as mean bias and rms.

I offer my wish list for additional figures that might serve well to enhance this manuscript.

–AMOC circulation streamfunction. Alhtough not directly observable, it is rather commonly shown in papers like this and it greatly adds to modeler-speak communication.

–Global and indo-pac ocean heat transports. I am particularly interested in Southern Ocean transports given the rather different ACC transports found in the models documented here.

---

## Referee Comment (RC2) · Luke Van Roekel (Referee) · 16 Jul 2019

I found the manuscript by Malcom Roberts et al to be well organized, nicely written, and nice to read. It covered all the main components I would like to see addressed in a paper like this, the only major exception is it would have been interesting to see how the simulated Madden Julian Oscillation responds to resolution. In our experiments with E3SM, we see little to no improvement (and perhaps a slight worsening) of the MJO in our equivalent HH experiment. It would be interesting to see how the MJO responds to the various combinations tested here. I would encourage the authors to

think about including a short discussion of this important intraseasonal oscillation, but do not believe it is necessary for this manuscript to be published.

Overall, I had no major objections. My two more general concerns are related to reproducibility and the values of parameters chosen for the simulations.

As noted by Dr. Griffies, I too was troubled by the statement in the data availability section that makes it impossible to reproduce these experiments except by folks within Hadley Center. This is an unfortunate decision by the Hadley Center, but I also don't think this should or can prevent publication of this work.

Second, in a few places I felt it would be helpful to more thoroughly mention the role of the chosen GM bolus kappa parameter. In particular, at low resolution the Drake Transport and simulated antarctic circumpolar current will be strongly dependent on the chosen bolus kappa value. I think it is important for the authors to more clearly state the dependence in Section 3.6 for example. I believe you could judiciously choose your value of bolus kappa to minimize the change in ACC transport across the resolutions studied.

I recommend publication with minor revisions, including point 2 above and the following line specific corrections.

1) Near Line 50 you could also reference our soon to be submitted manuscript on using E3SM to explore resolution effects under the highresmip protocol

Caldwell, P and co-authors, 2019: The DOE E3SM coupled model version 1: Description and results at high resolution, in prep for JAMES

2) on page 6, numerous subscript formatting needed for W/m2

3) right above 25, there are two MLs, I assume one should be LM?

4) Near line 25, I would also cite this paper on the large polynyas seen in other models

https://journals.ametsoc.org/doi/full/10.1175/JCLI-D-16-0741.1

5) Near line 30, why not use iceSAT for both hemispheres? I believe ICESAT thickness is a preferred benchmark to PIOMASS volume in the sea ice community.

6) line 6 page 10 – need to say high resolution atmosphere.

7) Your descriptions of Figure 12 in text (pg L27) are not terribly clear to me, for example, by West North Pacific, is this the region directly above the dateline? So north just means north of the equator?

8) Pg 12, line 21, suggest moving this sentence before the figure 16 sentence to improve flow.

9) Line23 page 13 – Stephenson -> Stevenson

10) Figure 17 – I'm not sure this figure adds to the discussion. As you cite (Stevenson et al and Wittenburg et al) a much longer simulation is required to appropriately resolve the NINO34 spectra. Further, at least to my eye, all simulations reproduce the HadISST spectrum fairly well. I would consider dropping this figure but leaving the discussion about observed variability. The figure only confirms what is seen in previous literature.

11) Broad comment about the conclusions, it would be helpful to include references to figures when you discuss biases again.

12) Page 14 L8 – do you have references to support the "Based on previous work"?

13) in data availability I would suggest changing the link to the CICE code, our oceans11 server is going away soon. I would point people to the CICE consortium page https://github.com/CICE-Consortium.

14) Bias figures would benefit from a summary statistic on panels (similar to Figure 4).

---

## Referee Comment (RC3) · Justin Small (Referee) · 21 Jul 2019

The paper describes a major contribution to the HighResMIP experiments, namely a very comprehensive suite of resolution comparisons by the UK Met. Office group with their latest climate model. The HighResMIP protocol includes a 1950 spin-up (∼30 years), a 1950 control (100 years) and transient simulations of 1950-2050. (These choices were governed by necessity: it is too expensive to run high-resolution simulations over many 100s of years as in main CMIP6 protocol.)
This initial paper focusses on the first two components with an aim to determine whether the protocol is effective. For example, is the spin-up too short, is the upper ocean in control still drifting rapidly after 100 years (not ideal for comparisons with transient), is 1950 a good base point? Most of these questions are comprehensively addressed except for the latter (appropriateness of 1950) which could be addressed in future work on transient simulations.

The paper is thorough: the set of experiments is outstanding in its breadth of resolutions for climate models: and the paper should be a good reference for the community and HighResMIP users in particular. I only have minor comments before, in my opinion, it should be published.

Comments

I think the main question is whether these simulations make robust controls against which transient simulations can be compared, and I think you do not address this directly (especially in the Conclusions and Abstract – the Abstract in fact does not clarify that you only look at the 1950 runs.) I would like a bit more discussion of this. Do you think it appropriate to identify climate change by subtracting the drift of your 1950 runs from the transient runs?

Page 3, line 6 – a little more detail on the atmosphere grid, e.g. how many levels in $\sim$ lowest 1km, how many levels in stratosphere. Line 8 – same for ocean, how deep does the 1m spacing go, # points in upper 100m and approximate spacing in main thermocline? This could go in the Table.

Line 15. Presumably MACv2-SP scheme is used in both control and transient simulations? Line 20. For the unfamiliar – what is the "USSP launch factor"??

Page 4, lines6-7 is a repeat re aerosols.

Line 11. Re solar cycle: do you expect the solar cycle to have a major impact, thus requiring your protocol of smoothing out the solar cycle?

Table 1 – a curious point, why is CMIP6 nominal resolution for atmosphere ∼ 2*grid spacing, but for ocean it is ∼ 1*grid spacing? Or do I misunderstand? Also, put a statement in the text that you use the word resolution to mean "grid spacing" if that is what you do (in common with most papers).

Also, add to Table whether runs are spun up or initialized from another run, then add total run length.

Page 5 lines 25-26. It is impressive that LL, MM and MH are run for extended long periods which helps put the 100 year results in context.

Page 6, line 14-16. Add units. Lines 13 to 16 could be usefully included in a Table and combined with the coupled model values.

Line 24. ML is repeated twice.

Line 29. Parentheses around "(beyond . . . model)"

Line 32. I'm not an expert on this, but I've heard that standard resolution PI controls are typically tuned so that the TOA imbalance « 0.1W/m2. Your values are somewhat larger – any comment?

Page 7. Line 5. Delete "in" before "near"

Line 11. I would say the reduction of SW CRF bias off North America is notable smaller than other regions.

Fig. 7. There is a linear feature in Figs 7a-c in Southern Hemisphere at about the latitude of south-west tip of Australia. Is this an artefact of interpolation, or in original EN4 products?

Page 8 line 8 – cold bias possibly due to "the experimental design of using EN4" initial conditions. Can you expand on this? I remember early versions of CESM2 also had a cold bias for some runs initialized from Levitus. Is there something about these models that lead the surface to cool when initialized from observations?

Line 12-13. What about the typical warm bias of many degrees seen off the coast of N America or Japan due to western boundary current separation problems – do you see them in LL, and do they reduce at higher resolutions?

Line 28, "particularly in the ocean upwelling regions" – you could reference Gent et al 2010 (Clim. Dyn.), Small et al 2014 (JAMES), 2015 (J. Clim) who found consistent results in CCSM4,CESM1 regarding reduction of SST bias with atmosphere resolution.

Line 29-30. This is also consistent with CESM e.g Small et al 2019, Climate Dynamics (2019) 52:2067–2089, their Fig. 9 – high resolution cools at the coast (reducing bias) but warms further offshore. In general are Figs. 7i-k consistent with Griffies et al 2015, von Storch et al 2016( Ocean Modelling, 108, 1-19)? See also later.

Fig. 7d. The changes off Peru-Chile are smaller than I would expect from Figs 7a,b. Any thoughts? Does it relate to interpolating to a common, coarse grid?

Fig. 9. It seems that surface temperature over Greenland improves, but less dramatically than over other parts of Arctic. Is the bias over Greenland a true model problem, or lack of observational data? Is there an ice-sheet component to the model?

Section 3.3 illustrates generally large changes with resolution. The depth scale in Figs 10, 11 is strange, probably stretched too much in upper ocean. Also, why not show HH?

Griffies et al 2015 show some role for submesoscale (parameterization) in the heat budget. Does your model have such a parameterization? In Small et al 2014 we speculate that lack of submesoscale param. in the high-res model might explain some differences with the standard resolution model, which did contain the parameterization.

To complement Figs 10, 11, I think it is very useful to see spatial maps of temperature and salinity at say 500m or 1000m, at end of 100 year run, to look at regional detail. For example, do problems with Mediterranean Outflow, or Agulhas leakage, contribute to bias and drift?

[Figure]

Page 10, line 30. I think this is a common problem with low resolution models, papers by e.g. I. Richter discuss this at length.

Figure 12. It is interesting that changes due to ocean resolution (Figs 12i,j) are comparable in magnitude to those due to atmosphere resolution.

Section 3.6. High resolution CESM also had a weaker ACC transport than standard resolution CESM (Small et al. 2014). Any thoughts why HH, MH has weaker ACC than LL (in addition to your explanation for MM)?

Fig. 17. All the power spectra look quite sensible, but then I noticed the log scale ordinate. If plotted with linear ordinate would it be easier to see model differences and model biases?

Section 3.5. I think you should emphasize more how good the high-resolution models (MH, HH) are in the deep ocean in terms of AMOC mean profile. Put this in the context of what the AMOC actually represents in terms of major ocean currents.

Fig. 18, 19. Consider adding contours of Sea Level Pressure for the composites.

Section 3.7. Also, consider the paper: Deser et al 2017, J. Clim. "The Northern Hemisphere Extratropical Atmospheric Circulation Response to ENSO: How Well Do We Know It and How Do We Evaluate Models Accordingly?"

Finally, there has been a recent paper published (which unfortunately I cannot find now, but I think was published in 2019) that showed the slightly surprising result that although a high-resolution ocean model gave much reduced SST bias in the N. Atlantic in the first 50 years of the run, compared to low-res, the biases looked much more similar (between resolution) at the end of a multi-century integration. (In other words, the high-res bias increased substantially with time). Their paper used forced ocean-ice models. I wonder if this has relevance for your paper which only looks at 100 years of high-res. Perhaps the results will differ between coupled and forced simulations.

---

## Author Response (AR1)

[revised manuscript text omitted]
, 2012). PIOMAS has been shown to compare well with ICESat thickness for the periods where ICESat is present (Schweiger et al. 2011), and this gives us confidence to use the data throughout the year and over the longer evaluation period 1990-2009. The seasonal cycle amplitude and phase of sea ice area is well captured in the models except for LL in the Arctic which has too much sea ice. All the models have more sea ice volume than is indicated by the PIOMAS model and ICESat in the Arctic and Antarctic respectively. In the Arctic the volume increases over time in

10  the MM and MH simulations while reducing somewhat in LL, while in the Antarctic the MM volume starts lower than the other models but adjusts to a similar mean state.

**3.2 SST adjustment and biases**

The SST biases at the end of the control-1950 period (averaged over years 50-100) are shown in Fig. 7 for each model resolution. These are shown both as model bias compared to initial condition (top row), and as inter-resolution differences, for

15  the total change (due to atmosphere+ocean resolution), change due to atmosphere resolution alone, and change due to ocean resolution alone (rows two, three, four respectively). A common feature across the models is a generally cold mid-latitude bias, which may partly reflect the experimental design of using EN4 1950-54 initial conditions, the short spinup-1950 period, the constant 1950's forcing derived from CMIP6 and a consequently negative TOA in the first few decades (Fig. 2a), but is also a feature found in many climate models and experiments (see Flato et al. 2013, Fig. 9.13; Kuhlbrodt et al. 2018).

20  The biases typical for a 100 km ocean model (Danabasoglu et al. 2014) are evident in Fig 7(a) for LL, and also described in Kuhlbrodt et al. (2018). They are strongest over the boundary currents in the North Atlantic and North West Pacific, with cold biases of more than 5K, and over the tropics with a cold bias of 1-2K. Warm biases over the stratocumulus decks to the west of Southern Africa, South America and California are also evident, as well as warm biases in the regions where boundary currents separate from the coastline (particularly Gulf Stream and Kuroshio). 
[revised manuscript text omitted]

Given the experimental design, it is difficult to disentangle drifts due to imbalances in surface forcing from processes that may be poorly represented. As discussed in Griffies et al. (2015), von Storch et al. (2016) and Small et al. (2019), the evolution of the subsurface ocean depends on fluxes of heat (and salt) from either parameterised or explicitly represented processes (e.g. vertical diffusion, advective fluxes from mean, mesoscale and sub-mesoscale circulation). The changes in the subsurface ocean over time with resolution shown here are consistent with these previous studies, which would indicate an important role for upward heat transport from vertical mesoscale fluxes. The models shown here do not contain a submesoscale eddy parameterisation, which may be important to represent unresolved processes (Fox-Kemper et al. 2011), but they do have enhanced ocean vertical resolution in the near surface (Table 1) compared to the models referenced above which may also be important.

**Commented [RM19]:** 16

**Commented [RM20]:** 47

The above has shown that 100 years is insufficient to saturate the deep ocean drifts, as would be expected. However, some differences with resolution do seem to be robust and possibly linked to process improvement, particularly in the Southern Ocean where the eddy-rich MH simulation greatly reduces the deep warm temperature drift, and in the North Atlantic for both temperature and salinity.

The spatial patterns of the temperature and salinity biases at 950m depth are shown in Fig. 12 for LL, MM, HH, as a mean over years 50-100, compared to EN4 1950-54. As indicated in the previous figures, the LL model has warming and increased salinity over much of the Atlantic with an enhancement in the western tropical Atlantic. There biases are mostly reduced in MM, HH simulations, where the main bias switches to the eastern North Atlantic with potentially some role for the Mediterranean outflow. In the region of the Gulf Stream separation from the North American coast, the HH model has a cold and slightly fresh bias opposite to the lower resolutions. Over the rest of the global oceans, the LL model has warming and increase in salinity at mid-high latitudes in the Pacific Ocean which is somewhat reduced in HH, while all the resolutions have cooling and freshening in the north west Indian Ocean and a warming in the south west Indian Ocean.

[revised manuscript text omitted]

The depth profile of AMOC (Fig. 15b) indicates a strengthening at most depths with increased ocean resolution, with all
5  models having a maximum at around 1000 m consistent with the observations. The shape of the AMOC profile at depth in MH/HH agrees far better with the observations (a difference of 5 Sv at 3000m between LL and HH), which is likely to be important for global deep water masses and circulation. However, as discussed above, the peak transport at 1000 m becomes considerably stronger than observations in HH. The seasonal cycle of AMOC with annual mean removed (Fig. 15c) indicates the higher resolution models can match the observed magnitude and phase. The AMOC minimum in March corresponds to the
10  period of maximum variance, with reduced variability in summer.

The spatial structure of the AMOC in both depth and potential density (referenced to 2000m) in shown in Fig. 16. The strength of the AMOC in depth space and the depth of the return flow both increase with resolution over all latitudes apart from the northern North Atlantic (50-60°N) where LL is marginally stronger. The depth of the southward flow deepens at higher resolutions (as seen in Fig. 15b), and at 30°S the AMOC is 3-5 Sv stronger at higher resolution. In density space the AMOC
15  is stronger everywhere at higher resolution, with clearer indication of enhanced exchange with the Nordic Seas north of 60°N as well as increased subpolar strength. The subtropical cell at 20-30°N at $\sigma_{2000}$=34 also becomes enhanced, consistent with stronger North Atlantic Current transport. These changes are fairly typical of high resolution model simulations (Hirschi et al. 2019), though not all models produce a stronger AMOC at higher resolution (Sein et al. 2018).

The NHT dependence on latitude is shown in Fig. 17 for the global, North Atlantic and Indo-Pacific basins, where the
20  individual components (total, resolved advective components, diffusion and parameterised eddy-induced velocity) are also indicated. In the North Atlantic (Fig. 17b), the eddying advection component for LL is only visible near the equator, while the eddy-induced transport associated with the Gent-McWilliams parameterisation reaches ~0.1 PW around 40°N. The MM and HH models have similar eddying advective components, but the eddy-rich ocean has considerably stronger mean transport which better agrees with observations between the equator and 40°N. One aspect of note is the increased HH NHT northwards
25  of 45°N towards the Arctic as also seen in Roberts et al. (2016). It is unclear if this is excessive compared to observations, but if so it would imply that the ocean does not lose enough heat to the atmosphere at these latitudes.

For the global ocean (Fig. 17a), the higher resolutions have noticeably enhanced poleward transport at higher latitudes. In the Southern Ocean around 40°S, the balance of components is somewhat different in LL, where the resolved time-mean advection term is considerably more positive than the mean advection of the higher resolutions. In LL this is compensated by both the
30  parameterised eddy-induced advection (EIV) and the diffusive term (which also includes a component from the eddy parameterisation), while for higher resolutions only the eddying advection compensates, with the result that the southward transport of heat in LL is smaller. In the Indo-Pacific the total transports have smaller differences, though at higher resolutions this is due to a stronger compensation between time-mean and eddying advective components.

**Commented [RM26]:** 53

**Commented [RM27]:** 6

**Commented [RM28]:** 7

**3.6 Antarctic Circumpolar Current**

The time evolution of the Antarctic Circumpolar Current (ACC) transport, calculated as the volume transport through Drake Passage, is shown in Fig. 18. The mean net eastward transports of 155, 90 and 125 Sv respectively for LL, MM and MH models compares to the recent observational range of 173±11 Sv (Donohue et al. 2016), with earlier estimates lacking a robust

5  barotropic component (e.g. 137±8 Sv; Cunningham et al. 2003). Using the former measure, LL is closest to the observational range, while MM is only 40% of it. Figure 18 indicates that the impact of different atmosphere resolutions is small compared to the impact of ocean resolution. A part of the deficit in the M ocean model is due to a strong counter-current around the Antarctic shelf of about 20 Sv, together with changes to the density front, as discussed more fully in Menary et al. (2018) and Kuhlbrodt et al. (2018). The ACC in MH and HH remains lower than observed (as also seen in Small et al. 2014) – these have

10  negligible counter-currents, but perhaps they still have too much southward heat transport (Fig. 17a) and consequently weakened density gradient, and the H ocean resolution is still only marginally eddy-resolving at these latitudes (Hallberg et al. 2013). Despite a reduced transport, however, the frontal structures associated with the ACC, some with a barotropic structure, are much more evident in the M and H ocean models (not shown).

It has been shown (e.g. Kuhlbrodt et al. 2012) that the value of a constant coefficient for eddy parameterisation can influence

15  the ACC transport via the meridional density gradient, and it is possible that the varying coefficient used in LL (Table 2) may play a similar role. Use of such a scheme in the higher resolution models may well increase the ACC transport for similar reasons, at the expense of removing explicitly resolved mesoscale processes.

**3.7 ENSO and MJO variability**

As the dominant mode of interannual tropical variability, El Niño-Southern Oscillation (ENSO) is a key aspect of climate

20  variability with worldwide impacts (Timmermann et al. 2018). Over time there has been some improvement in modelling ENSO in global climate models (e.g. Bellenger et al. 2014), with HadGEM3-GC3.1 performance described in Williams et al. (2017), Kuhlbrodt et al. (2018) and Menary et al. (2018). Fig. 19 shows the power spectrum of Niño3.4 monthly surface temperature anomalies, calculated using a periodogram method with 50 years of data in each sample, and a 25 year overlap between samples, with the average power spectrum and range (shading) shown. Only the LL and MM models are shown since,

25  as indicated in Wittenberg (2009) and Stevenson et al (2010), 100 years is insufficient for a robust spectrum. The mean spectra from these models agrees well with that from HadISST1.1 observations for 1877-2018 (Rayner et al. 2003), with the other resolutions having little obvious difference (not shown). The standard deviations of the mean Nino3.4 DJF SST from the models are all slightly higher than the observed value of 0.93.

The composite December-January-February (DJF) mean surface temperature patterns relating to El Niño and La Niña events

30  are shown in Fig. 20, with events defined when the DJF Niño3.4 index exceeds ±0.7 K. There is a robust pattern to the global surface temperature anomalies which agree well (over the ocean) with the observed HadISST1.1 dataset and over the land with the Climate Research Unit time series 4.01 data set (CRU TS; Harris et al., 2014) 2 m temperatures for 1901-2016 (Fig. 20f).

**Commented [RM29]:** 18

**Commented [RM30]:** 51

**Commented [RM31]:** 10

**Commented [RM32]:** 20

The extension of the El Niño and La Niña patterns past the dateline is slightly excessive in the LL model, which is a common bias (e.g. Guilyardi, 2006; Roberts CD et al., 2018). The teleconnections to land surface temperature anomalies are robust over the Americas and Africa, but less so over Eurasia; the models with H atmosphere tend to have stronger negative anomalies over Northern Europe with El Niño, but these time series are shorter and hence have far fewer events. The surface pressure anomalies from the models (contours in Figs. 20, 21) are also consistent with the observations from HadSLP2 (Allan and Ansell, 2006).

The equivalent composite rainfall patterns are shown in Fig. 21 for the models and GPCP2.3 observations for 1979-2014 (Adler et al. 2018) for El Niño and La Niña events. The extension of the SST pattern into the western Pacific in LL is also evident here as excessive precipitation at the equator in the West Pacific, with some improvement at higher resolutions. All model resolutions mirror the observed teleconnections quite faithfully, though the dry anomaly with El Niño events over South Africa is not robustly captured.

The near 1:1 ratio of El Niño to La Niña events found in observations is replicated in the models, but the ratio of Cold Tongue (CT, East Pacific) to Warm Pool (WP, Central Pacific) events, as defined by the indices in Ren and Jin (2011), is less well represented, as noted in Fig. 20 titles. The LL model has a near 1:1 ratio of such events compared to the observed 2:1, the higher resolution ocean models have far fewer WP events compared to CT but there seems to be little systematic change with resolution.

There seems to be only minor differences in the ENSO performance in the models at different resolutions, mainly in slight differences to the SST composite. While 100 years is not long enough to assess the power spectrum, as noted previously by Wittenberg (2009) and Stevenson et al (2010), and ENSO composites based on relatively few events can be uncertain due to internal variability (Deser et al. 2017), the composite patterns of surface temperature and precipitation show relative robustness.

The Madden-Julian Oscillation (MJO) dominates the tropical intraseasonal variability (Madden and Julian, 1971) and is characterized by eastward propagation of deep convective structures moving along the equator with an average phase speed of around 5 m/s with periods between 30 and 90 days, together with other modes (Wheeler and Kiladis, 1999). The symmetric wave spectra, expressed as the ratio of raw power of outgoing longwave radiation (OLR) and a smoothed background spectra that highlights the major equatorial wave modes and their dispersion relationships compared to that of a shallow water model (represented as lines) is shown in Fig. 22a from daily NOAA observations (Lee 2014). As shown in Menary et al. (2018) and Williams et al. (2017), the HadGEM3-GC3.1 coupled model underestimates the power in the MJO mode (wavenumbers 1-3 and periods 30-90 days) and Kelvin mode (Fig. 22b,c,d), though for the latter there is a marginal increase in power at higher resolutions. None of the resolutions are able to produce inertia-gravity waves (IG), or mixed Rossby-gravity waves in the anti-symmetric spectrum (not shown).

Commented [RM33]: 19

Commented [RM34]: 55

Commented [RM35]: 8

**4 Summary and discussion**

As part of the CMIP6 HighResMIP project, a wide range of coupled model simulations with atmosphere resolutions between 250 km and 50 km, and ocean resolutions from 100km to 8km, have been performed with the HadGEM3-GC3.1 model. We have shown that increased model resolution in the atmosphere and ocean can have considerable impact on climate model biases of the mean state and variability, both at the surface in terms of temperature and precipitation (Figs 4, 7, 9, 13), as well as in the deeper ocean (Figs. 10-12).

We have demonstrated that the new CMIP6 HighResMIP experimental design, with only a multi-decadal spinup and 100 year simulation length, is sufficiently long to robustly establish some of these responses in model bias (Fig. 8). This has enabled the use an eddy-rich 8 km ocean model within the same suite of experiments, to make a more comprehensive chain of resolutions, and hence further test the robustness of our results. These experiments may also enable better understanding of the model adjustment process (so-called spinup from initial conditions), which tends not to be a focus of the standard CMIP simulations with a long pre-industrial spinup. This makes it harder to understand why the deep ocean adjustment process timescales may be different with different resolutions, and what role these biases might play in model sensitivity to changes in forcing.

We find that increased ocean resolution is key to reducing many of the most common SST biases (Fig. 7), while a combination of ocean and atmosphere resolution significantly improves the large tropical Atlantic precipitation biases seen in typical CMIP-resolution models (Fig. 13), the latter having the potential to cause considerable uncertainty in projections of future rainfall changes.

We have also found some potential links between the biases and the evolution of the mean state. Based on previous work (Jackson et al. 2015), it seems likely that the strengthened Atlantic Meridional Overturning Circulation (AMOC) and northward heat transport in the tropical Atlantic is linked to the improved SST biases and reduced precipitation (and ITCZ) biases, which in turn may be associated with some of the deeper ocean biases. These may also link to the different spinup behaviours seen in the different models. The initial drop in AMOC and Northward Heat Transport in the LL model (Fig. 14) causes a cooling in the North Atlantic and Arctic, with a consequent increase in sea ice. Over time the stronger temperature (and salinity) contrast between equator and pole drives an increase in AMOC which gradually warms the Arctic. This increase in AMOC could be enhanced by the increased tropical Atlantic salinity bias in LL, which would increase the density of water reaching the northern North Atlantic and enable a stronger AMOC circulation to develop. Using the same initial conditions and short spinup in all experiments may enable better understanding of such adjustment processes than is generally possible in standard CMIP simulations.

[revised manuscript text omitted]

With thanks to Paul Earnshaw, Dan Copsey, Matthew Mizielinski, Tim Graham and many other colleagues at the Met Office
20 for help in various aspects of model simulation, data processing and analysis tools.

We would also like to thank the reviewers Stephen Griffies, Luke van Roekel and Justin Small, and the editor, for their help in improving this manuscript.

**Tables**

| Atmos. model level | Atmos. height (km, approx.) | Ocean model level | Ocean depth (m) |
|---|---|---|---|
| 1 | 0.02 | 1 | 0.5 |
| 10 | 0.8 | 10 | 14 |
| 20 | 1.7 | 20 | 61 |
| 30 | 4.5 | 30 | 180 |
| 50 | 14.1 | 50 | 1387 |
| 85 | 85 | 75 | 5902 |

5    Table 1: A selection of model levels in the atmosphere and ocean and their corresponding height/depth.

| Ocean grid | ORCA1 | | ORCA025 | | ORCA12 | |
|---|---|---|---|---|---|---|
| Ocean timestep (min) | 30 | | 20 | | 5 | |
| Gent-McWilliams; GM coeff; isopycnal diff | On; variable (Held and Larichev, 1996); 1000 | | Off; 0; 150 | | Off; 0; 125 | |
| Momentum dissipation | Laplacian; $20 \times 10^3$ m$^2$s$^{-1}$ | | Bilaplacian; $-1.5 \times 10^{11}$ m$^4$s$^{-1}$ | | Bilaplacian; $-1.25 \times 10^{10}$ m$^4$s$^{-1}$ | |
| Snow on sea ice albedo: near-infrared; visible | 0.68; 0.96 | | 0.7; 0.98 | | 0.7; 0.98 | |
| Atmos timestep (min) | 20 | 15 | 15 | 10 | 15 | 10 |
| USSP launch factor (for QBO) | 1.3 | 1.2 | 1.2 | 1.2 | 1.2 | 1.2 |
| Experiments submitted to CMIP6 ESGF | spinup-1950; control-1950, hist-1950 | spinup-1950 control-1950 | spinup-1950; control-1950, hist-1950 | control-1950, hist-1950 | spinup-1950; control-1950 | control-1950, hist-1950 |
| Atmos model name (mid-latitude grid spacing) | N96 (135 km) | N216 (60 km) | N216 (60 km) | N512 (25 km) | N216 (60 km) | N512 (25 km) |
| CMIP6 nominal resolution (atmosphere, ocean) | 250 km, 100 km | 100 km, 100 km | 100 km, 25 km | 50 km, 25 km | 100 km, 8 km | 50 km, 8 km |
| HadGEM3-GC31 naming convention | **LL** | **ML** | **MM** | **HM** | **MH** | **HH** |

Table 2: Parameter value changes between different model ocean (from top) and atmosphere (from bottom) resolutions, together with resolution naming conventions.

| Model name | CMIP6 resolution (atmos-ocean) km | Initial condition | Total years (spinup years) | Nodes (atmos-ocean) | Max turnaround (years per day) | Output per year (TB) |
|---|---|---|---|---|---|---|
| LL | 250-100 | LL-spinup (30 years) | 1130 (30) | 12-2 | 4 | 0.13 |
| MM | 100-25 | MM-spinup (30 years) | 680 (30) | 50-24 | 1.3 | 0.73 |
| HM | 50-25 | MM-spinup (30 years) | 117 (0) | 90-24 | 0.5 | 2.8 |
| MH | 100-8 | MH-spinup (30 years) | 205 (30) | 34-171 | 0.45 | 2.0 |
| HH | 50-8 | MH-spinup (30 years) | 100 (0) | 90-171 | 0.4 | 4.5 |

Table 3: For each model resolution for the control-1950 simulation, the nominal CMIP6 resolution, the initial condition, total simulated years, and costs of various model resolutions on a Cray XC40 with 36 cores per node, together with raw model output volumes.

**Figures**

**CMIP6 HighResMIP simulations**
Physical model only x 2 resolutions, simplified aerosol optical properties (MACv2-SP) recommended

[Figure]

**Atmosphere-land-only**, 1950-2014 (→ 2050)
Forced by observed SST and sea-ice and historic forcings (→ projected)
highresSST-present (→ highresSST-future)

| 1950 | Historic forcings | 2014 | Future forcings | 2050 |
|---|---|---|---|---|
| | *highresSST-present* | | *highresSST-future* | |

**Coupled climate**, 1950-2014 (→ 2050)
Forced by constant 1950 and historic forcings (→ projected)
Initial coupled spin-up period ~ 30-50 years from 1950 EN4 ocean climatology
spinup-1950, control-1950, hist-1950
(→ highres-future, future-2050)

Future projected forcing
2015-2050, *highres-future*

Constant 2050's forcing
*future-2050*

2050

Historic 1950-2014 forcing
*hist-1950*

2014

1950

1950

Constant 1950's forcing
*spinup-1950*

Constant 1950's forcing
*control-1950*

Optional extension

[revised manuscript text omitted]

5    switched off. For Global and Indo-Pacific: the black circles and lines are observational estimates from Ganachaud and Wunsch (2003) with uncertainty. For the Atlantic: the black circles and lines are observational estimates from Ganachaud and Wunsch (2003) with uncertainty; the black triangle is RAPID-MOCHA (Johns et al. 2011); white diamond - Bryden and Imawaki (2001); gray circle - Talley (2003); white open circle - Lumpkin and Speer (2007); white inverted triangle - McDonagh et al. (2010); white star - Lozier et al. (2019). The coloured circles in (b) at 26.5°N correspond to the mean values using the

10   RapidMoc calculation as in Fig. 13.

[Figure]

Figure 18: Time series of the Antarctic Circumpolar Current (ACC) transport calculated between Antarctica and South America across Drake Passage for spinup-1950 and control-1950 simulations.

[Figure]

Figure 19: Power spectrum of Niño3.4 surface monthly temperature variability for LL, MM resolution models and HadISST1.1 observations. The coloured line shows the mean power spectrum for all years, the dashed line shows the first 50 years, while the shading shows the range over all 50 year sub-sampled periods. The number in brackets is the standard deviation of the DJF mean SST in the Niño3.4 region.

[Figure]

Figure 20: Composite DJF mean surface temperature (colours) for (top) El Niño and (bottom) La Niña from models and observations based on El Niño and La Niña events. The number of events sampled is shown in the title, and the proportion of El Niño events classified as Cold Tongue (CT) and warm pool (WP). Events are defined as years when the Niño3.4 DJF SST anomaly exceeds 0.7 K. The observations are a combination of HadISST1.1 over the ocean and CRU 2 m temperatures over land, with values masked over HadISST1.1 sea ice regions. Mean sea level pressure anomalies are shown as contours with interval 0.5 hPa, with observations from HadSLP2.

[Figure]

5  Figure 21: As Fig. 18 but for DJF composites of mean precipitation (top) El Niño and (bottom) La Niña from models and GPCP2.3 precipitation, with mean sea level pressure anomalies as contours. The number of events sampled is shown in the title, and the proportion of El Niño events classified as Cold Tongue (CT) and warm pool (WP). Events are defined as years when the Niño3.4 DJF SST anomaly exceeds 0.7 K.

[Figure]

Figure 22: (a) The ratio between raw power spectrum of outgoing longwave radiation (OLR) and a background spectrum from daily NOAA observations (1989–2008) averaged over 15°S-15°N. (b) – (d) Bias between HadGEM3-GC31-LL/MM/HH and observations respectively, for model years 2000-2020. All values are log of OLR power. See Wheeler and Kiladis (1999) for details. CPD = Cycles per day.

"Description of the resolution hierarchy of the global coupled HadGEM3-GC3.1 model as used in CMIP6 HighResMIP experiments" by Malcolm J. Roberts et al.

Responses to reviewers

Reviewer 1: Stephen Griffies
My main concern is with the following statement on page 15, lines 9-10:
"Due to intellectual property rights restrictions, we cannot provide either the source code or documentation papers for the UM 10 or JULES." That is a very unsatisfying situation that, in effect, means these simulations are not reproducible by any willing and able person outside of the Hadley Centre. I am disappointed that the Hadley Centre has failed to fully embrace scientific programming of the 21st century where open development / open access greatly supports the climate science endeavour by enabling reproducibility.
**Response 1**:
The initial wording of this model code availability was poorly chosen, and has been clarified in the Data and code availability section. The model codes are available to use as now stated.
I recommend publication with the following minor points.
page 6 lines 14 and 15: please give units
**Response 2**: Done

page 6: line 28: please add following citation along with Griffies et al 2015:
Preconditioning of the Weddell Sea polynya by the ocean mesoscale and dense water overflows, 2017: C.O. Dufour, A.K. Morrison, S.M. Griffies, I. Frenger, H.M. Zanowski, M. Winton, Journal of Climate, vol. 30, 7719-7737, doi:10.1175/JCLI-D-16-0586.1
**Response 3**: Done

page 10, paragraph with line 15: Is worth noting that interior biases can be exacerbated by spurious mixing, especially at the 1/4 ocean resolution. Citations to work from Alex Megan as well as Griffies, Illicak, others can be made.
**Response 4**: Done, Page 11, line 16.

Figures 7,8,12 could be more useful with statistical information such as mean bias and rms.
**Response 5**: Done for Figures 7,12. Figure 8 is simply the difference between start and end periods, so I don't think this is useful here.

I offer my wish list for additional figures that might serve well to enhance this manuscript.
–AMOC circulation streamfunction. Although not directly observable, it is rather commonly shown in papers like this and it greatly adds to modeler-speak communication.
**Response 6**: New Figure 16 showing the AMOC in both depth and density space, with text on Page 13, line 11.

–Global and indo-pac ocean heat transports. I am particularly interested in Southern Ocean transports given the rather different ACC transports found in the models documented here.
**Response 7**: New Figure 17 now shows the northward heat transport for global, Atlantic and Indo-Pacific basins, with some extra breakdown in components. Text included on Page 13, line 19 onwards.

Reviewer 2: Luke Van Roekel (Referee)

It covered all the main components I would like to see addressed in a paper like this, the only major exception is it would have been interesting to see how the simulated Madden Julian Oscillation responds to resolution. In our experiments with E3SM, we see little to no improvement (and perhaps a slight worsening) of the MJO in our equivalent HH experiment. It would be interesting to see how the MJO responds to the various combinations tested here. I would encourage the authors to think about including a short discussion of this important intraseasonal oscillation, but do not believe it is necessary for this manuscript to be published.

**Response 8**: An additional Figure 22 has been included to show the wavenumber/frequency spectra in the tropics from models and observations, which includes a component of the MJO. Additional text has been included on page 15, line 22.

Overall, I had no major objections. My two more general concerns are related to reproducibility and the values of parameters chosen for the simulations. As noted by Dr. Griffies, I too was troubled by the statement in the data availability section that makes it impossible to reproduce these experiments except by folks within Hadley Center. This is an unfortunate decision by the Hadley Center, but I also don't think this should or can prevent publication of this work.

**Response 9**: See above response to Reviewer 1, the text has been clarified.

Second, in a few places I felt it would be helpful to more thoroughly mention the role of the chosen GM bolus kappa parameter. In particular, at low resolution the Drake Transport and simulated antarctic circumpolar current will be strongly dependent on the chosen bolus kappa value. I think it is important for the authors to more clearly state the dependence in Section 3.6 for example. I believe you could judiciously choose your value of bolus kappa to minimize the change in ACC transport across the resolutions studied.

**Response 10**: The dependence of the Drake Passage transport on this kappa parameter indeed has been shown (Kuhlbrodt et al. 2012), but only for models that have a single fixed scalar value for the eddy-induced diffusivity. In N96ORCA1 however, this parameter is calculated at every time step and at every grid column as a function of the vertical density gradient (Kuhlbrodt et al. 2018, Storkey et al. 2018). In this case the relationship to DP transport is much less clear. I have added some text to the end of Section 3.6 to reflect this comment.

1) Near Line 50 you could also reference our soon to be submitted manuscript on using E3SM to explore resolution effects under the highresmip protocol Caldwell, P and co-authors, 2019: The DOE E3SM coupled model version 1: Description and results at high resolution, in prep for JAMES.

**Response 11**: Done, page 2, line 16, this is included though I need the full author list for the reference.

2) on page 6, numerous subscript formatting needed for W/m2

**Response 12**: Done

3) right above 25, there are two MLs, I assume one should be LM?

**Response 13**: Done

4) Near line 25, I would also cite this paper on the large polynyas seen in other models
https://journals.ametsoc.org/doi/full/10.1175/JCLI-D-16-0741.1

**Response 14**: Done

5) Near line 30, why not use iceSAT for both hemispheres? I believe ICESAT thickness is a preferred benchmark to PIOMASS volume in the sea ice community.

**Response 15**:

The problem with ICESat is that it was only around from 2003-2008 and did not have
complete temporal coverage (the data is only available for parts of the year). As the laser onboard ICESat failed virtually from the onset, they had to use "campaign mode" and only turned on the laser for short periods every now and then (like focussed aircraft campaigns). So there isn't a huge amount of data and it doesn't get used much for these purposes (we've never seen anyone use it for large-scale climate model evaluation as here). PIOMAS has been shown to compare well with ICESat thickness for the periods where ICESat is present (Schweiger et al. 2011) and this gives us confidence ti use the data throughout the year and over the longer evaluation period 1990-2009.

However we should make clear that PIOMAS is used as a reference here rather than for direct validation. It is a good reference because it is well used and well understood.

We have added some text to page 8, line 5 to this effect.

6) line 6 page 10 – need to say high resolution atmosphere.

**Response 16**: Done

7) Your descriptions of Figure 12 in text (pg L27) are not terribly clear to me, for example, by West North Pacific, is this the region directly above the dateline? So north just means north of the equator?

**Response 17**: I have tried to make the text clearer, page 11 line 39.

8) Pg 12, line 21, suggest moving this sentence before the figure 16 sentence to improve flow.

**Response 18**: Done

9) Line23 page 13 – Stephenson -> Stevenson

**Response 19**: Done

10) Figure 17 – I'm not sure this figure adds to the discussion. As you cite (Stevenson et al and Wittenburg et al) a much longer simulation is required to appropriately resolve the NINO34 spectra. Further, at least to my eye, all simulations reproduce the HadISST spectrum fairly well. I would consider dropping this figure but leaving the discussion about observed variability. The figure only confirms what is seen in previous literature.

**Response 20**: I have changed the figure to just show the models that have enough years of simulation (LL, MM), and added to the text that other resolutions show little difference (given their shorter simulation length), page 14 line 24.

11) Broad comment about the conclusions, it would be helpful to include references to figures when you discuss biases again.

**Response 21**: I have added references to figures in the conclusions.

12) Page 14 L8 – do you have references to support the "Based on previous work"?

**Response 22**: Added Jackson et al. (2015).

13) in data availability I would suggest changing the link to the CICE code, our oceans11 server is going away soon. I would point people to the CICE consortium page https://github.com/CICE-Consortium.

**Response 23**: The code used within this modelling framework is now mentioned in this section, see response 1.

14) Bias figures would benefit from a summary statistic on panels (similar to Figure 4).
**Response 24**: Done (Figures 7, 12).

Reviewer 3: Justin Small
Comments
I think the main question is whether these simulations make robust controls against which transient simulations can be compared, and I think you do not address this directly (especially in the Conclusions and Abstract – the Abstract in fact does not clarify that you only look at the 1950 runs.) I would like a bit more discussion of this. Do you think it appropriate to identify climate change by subtracting the drift of your 1950 runs from the transient runs?

**Response 25**: The abstract has been modified to only mention the control simulations. Understanding whether we can identify a climate change signal with this experimental design is an open question but outside the scope of this work as indicated both by our questions (page 2 line 21) and on page 6, line 5. We have also added some discussion into the summary (page 17, line 8).

Page 3, line 6 – a little more detail on the atmosphere grid, e.g. how many levels in ∼ lowest 1km, how many levels in stratosphere. Line 8 – same for ocean, how deep does the 1m spacing go, # points in upper 100m and approximate spacing in main thermocline? This could go in the Table.

**Response 26**: New Table 1 now shows a selection of model levels and their associated height/depth.

Line 15. Presumably MACv2-SP scheme is used in both control and transient simulations? Line 20.
**Response27**: Clarified on page 3, line 18.

For the unfamiliar – what is the "USSP launch factor"??
**Response 28**: Clarified on page 3, line 26.

Page 4, lines6-7 is a repeat re aerosols.
**Response 29**: The description has been unified on page 3, lines 13-18.

Line 11. Re solar cycle: do you expect the solar cycle to have a major impact, thus requiring your protocol of smoothing out the solar cycle?
**Response 30**: it may well make no difference, but this was not tested so we were just extra careful.

Table 1 – a curious point, why is CMIP6 nominal resolution for atmosphere ∼ 2*grid spacing, but for ocean it is ∼ 1*grid spacing? Or do I misunderstand? Also, put a statement in the text that you use the word resolution to mean "grid spacing" if that is what you do (in common with most papers). Also, add to Table whether runs are spun up or initialized from another run, then add total run length.

**Response 31**: Done. Text on page 3, line 21 has clarified CMIP6 nominal resolution and defined model resolution. New Table 3 now included initial conditions and total length of simulations.

Page 5 lines 25-26. It is impressive that LL, MM and MH are run for extended long periods which helps put the 100 year results in context.
**Response 32**: It took a lot of time and CPU!

Page 6, line 14-16. Add units. Lines 13 to 16 could be usefully included in a Table and combined with the coupled model values.
**Response 33**: Done (now lines 21-22). Since the coupled values are shown in the Figure, and the atmosphere-only simulations are very different from the coupled model (given SST, sea-ice etc), I'm not sure that a table directly comparing these values would be informative.

Line 24. ML is repeated twice.
**Response 34**: Done (now page6, line 30)

Line 29. Parentheses around "(beyond . . . model)"
**Response 35**: Done, page 7 line 4

Line 32. I'm not an expert on this, but I've heard that standard resolution PI controls are typically tuned so that the TOA imbalance « 0.1W/m2. Your values are somewhat larger – any comment?
**Response 36**: This experiment is for 1950's conditions, and hence one would not expect or want a near-zero TOA. In addition, as in Menary et al. (2018), the TOA in a PI control can be greater than 0.1 and still be judged reasonable if the trend is negligible. For the HighResMIP experiment, with no extra tuning, we are pretty happy how low the TOA is, and we do not claim that the models are in equilibrium.

Page 7. Line 5. Delete "in" before "near"
**Response 37**: Done

Line 11. I would say the reduction of SW CRF bias off North America is notable smaller than other regions.
**Response 38**: page 7 line 19 we have qualified the North America change

Fig. 7. There is a linear feature in Figs 7a-c in Southern Hemisphere at about the latitude of south-west tip of Australia. Is this an artefact of interpolation, or in original EN4 products?
**Response 39**: I tested several methods of interpolating EN4 and models to grids and this did not change. I think it is simply that the isotherms line up similar to latitudes in this region and the models slightly shift to give a cooling further north.

Page 8 line 8 – cold bias possibly due to "the experimental design of using EN4" initial conditions. Can you expand on this? I remember early versions of CESM2 also had a cold bias for some runs initialized from Levitus. Is there something about these models that lead the surface to cool when initialized from observations?
**Response 40**: I added to Page 8, line 18 that the TOA is negative in the first few decades which would lead to a surface cooling. Without further analysis I don't have a better understanding of this.

Line 12-13. What about the typical warm bias of many degrees seen off the coast of N America or Japan due to western boundary current separation problems – do you see them in LL, and do they reduce at higher resolutions?

5 **Response 41**: I've added text to page 8 and the top of page 9 to mention these biases.

Line 28, "particularly in the ocean upwelling regions" – you could reference Gent et al 2010 (Clim. Dyn.), Small et al 2014 (JAMES), 2015 (J. Clim) who found consistent results in CCSM4,CESM1 regarding reduction of SST bias with atmosphere resolution.

10 **Response 42**: Done

Line 29-30. This is also consistent with CESM e.g Small et al 2019, Climate Dynamics (2019) 52:2067–2089, their Fig. 9 – high resolution cools at the coast (reducing bias) but warms further offshore. In general are Figs. 7i-k consistent with Griffies et al 2015, von Storch et al 2016( Ocean Modelling, 108, 1-19)? See also later.

15 **Response 43**: I have added the Small et al. and Griffies et al. references. Von Storch et al does not have an SST bias plot, so I've left that for the vertical diffusion discussion (your point below).

Fig. 7d. The changes off Peru-Chile are smaller than I would expect from Figs 7a,b. Any thoughts? Does it relate to interpolating to a common, coarse grid?

20 **Response 44**: I'm not sure I agree. The warm bias at the coast in LL reduces strongly to MM and disappears by HH, primarily due to atmosphere resolution, in line with what I would expect.

Fig. 9. It seems that surface temperature over Greenland improves, but less dramatically than over other parts of Arctic. Is the bias over Greenland a true model problem, or lack of observational data? Is there an ice-sheet
25 component to the model?
**Response 45**: There is no ice sheet model. It is difficult to say whether it is an observational or model problem. The different resolutions will represent the orography differently, and the representation of land ice is fairly simple.

30 Section 3.3 illustrates generally large changes with resolution. The depth scale in Figs 10, 11 is strange, probably stretched too much in upper ocean. Also, why not show HH?
**Response 46**: In these figures I only show the simulations with the same resolution as those with a corresponding spinup-1950. The vertical scale is chosen for clarity, it is meant to enhance the near surface, both because the differences are generally larger here and smaller at deeper levels, and to be proportional to the
35 model levels/depth spacing.

Griffies et al 2015 show some role for submesoscale (parameterization) in the heat budget. Does your model have such a parameterization? In Small et al 2014 we speculate that lack of submesoscale param. in the high-res model might explain some differences with the standard resolution model, which did contain the
40 parameterization.
**Response 47**: I have added a paragraph (page 10, line 30) to include some of this discussion.

To complement Figs 10, 11, I think it is very useful to see spatial maps of temperature and salinity at say 500m or 1000m, at end of 100 year run, to look at regional detail. For example, do problems with Mediterranean Outflow, or Agulhas leakage, contribute to bias and drift?

**Response 48**: An additional new figure 12 has been included showing temperature and salinity biases at 950m, and text on page 11, line 5.

Page 10, line 30. I think this is a common problem with low resolution models, papers by e.g. I. Richter discuss this at length.

**Response 49**: I added a sentence to reflect this, page 12 line 1.

Figure 12. It is interesting that changes due to ocean resolution (Figs 12i,j) are comparable in magnitude to those due to atmosphere resolution.

**Response 50**: I have noted that (page 12, line 8)

Section 3.6. High resolution CESM also had a weaker ACC transport than standard resolution CESM (Small et al. 2014). Any thoughts why HH, MH has weaker ACC than LL (in addition to your explanation for MM)?

**Response 51**: I have added some text – page 14, line 9.

Fig. 17. All the power spectra look quite sensible, but then I noticed the log scale ordinate. If plotted with linear ordinate would it be easier to see model differences and model biases?

**Response 52**: With a linear scale the plots become extremely noisy, as can be seen from the subsets of 50 year chunks. In addition I have removed the shorter simulations (responding to Reviewer 2).

Section 3.5. I think you should emphasize more how good the high-resolution models (MH, HH) are in the deep ocean in terms of AMOC mean profile. Put this in the context of what the AMOC actually represents in terms of major ocean currents.

**Response 53**: In have included an indication of this on Page 13, line 5.

Fig. 18, 19. Consider adding contours of Sea Level Pressure for the composites.

**Response 54**: Done.

Section 3.7. Also, consider the paper: Deser et al 2017, J. Clim. "The Northern Hemisphere Extratropical Atmospheric Circulation Response to ENSO: How Well Do We Know It and How Do We Evaluate Models Accordingly?"

**Response 55**: I have added a comment to this effect (page 15, line 20).

Finally, there has been a recent paper published (which unfortunately I cannot find now, but I think was published in 2019) that showed the slightly surprising result that although a high-resolution ocean model gave much reduced SST bias in the N. Atlantic in the first 50 years of the run, compared to low-res, the biases looked much more similar (between resolution) at the end of a multi-century integration. (In other words, the high-res bias increased substantially with time). Their paper used forced ocean-ice models. I wonder if this has relevance for your paper which only looks at 100 years of high-res. Perhaps the results will differ between coupled and forced simulations.

**Response 56**: Apologies, we could not figure out which paper this was referring to and hence have made no changes.